# GasketRAG: Systematic Alignment of Large Language Models with Retrievers

## Abstract

Retrieval-Augmented Generation (RAG) has emerged as a powerful method for enhancing the output quality of large language models (LLMs). However, existing retrievers are not specifically optimized for LLMs, and retraining them requires substantial resources. Furthermore, current approaches are often constrained to either improving the relevancy of retrieved documents or refining the documents post-retrieval. Various stages within the typical RAG pipeline present challenges in aligning LLMs with retrievers. To address these issues, we propose GasketRAG, a novel approach that introduces a gasket between the retriever and the LLM to improve their collaborative performance. By employing innovative techniques, we gather high-quality preference data and use the gasket to optimize both retrieval ranking and document refinement simultaneously. Our approach circumvents the need for constructing complex training and inference pipelines. In a fair comparison against the latest RAG methods across multiple test datasets, GasketRAG demonstrated a clear advantage. Our code and data are available anonymously at `https://anonymous.4open.science/r/9668`.

## 1 Introduction

Large language models (LLMs) often struggle with outdated knowledge, and updating them through retraining is both costly and inefficient. Retrieval-augmented generation (RAG) addresses this issue by retrieving passages relevant to a given query, allowing LLMs to incorporate up-to-date information and provide more accurate answers. RAG has demonstrated remarkable effectiveness across various NLP tasks (Yasunaga et al., 2023; Zhu et al., 2024b; Xiong et al., 2024; Xu et al., 2024a; Yue et al., 2024).

However, since the retriever and the LLM are typically trained separately, a disconnect exists between them, making it challenging for them to collaborate effectively (Ke et al., 2024). To be specific, retrievers are generally trained based on human preferences, designed to retrieve and rank documents in a way that aligns with human habits. However, the preferences of LLMs do not completely align with those of humans. Additionally, the documents or passages returned by retrievers often contain irrelevant information, referred to as noise. LLMs are highly sensitive to such noise (Xu et al., 2024c; Fang et al., 2024). Similarly, retrievers, constrained by their training data and model architecture, also exhibit different preferences when processing queries. Therefore, both retrievers and LLMs exhibit their own preference biases when dealing with human-written queries and documents. When integrated into the RAG pipeline, these biases affect the overall performance, a phenomenon we refer to as the *preference gap*. Tan et al. (2024) argue that LLMs tend to favor content they generate themselves over retrieved information, which highlights this gap. Existing work aimed at improving RAG performance generally focuses on either enhancing the retriever's ability to retrieve more relevant documents (Liao et al., 2024; Feng et al., 2024; Yoon et al., 2024a) or refining the retrieved documents to filter out the noise (Xu et al., 2024c; Qian et al., 2024; Yoon et al., 2024b). Moreover, some studies employ dynamic retrieval methods to balance LLM knowledge with retrieved information, aiming to generate more precise outputs (Asai et al., 2023; Xu et al., 2024b). However, this optimization of individual components in the RAG pipeline is based on human preferences, leveraging the human definition of "relevance" while overlooking the preferences of the retriever and LLM.

Narrowing this preference gap can help further improve the performance of the RAG system. We propose GasketRAG, which introduces an intermediate model called *gasket*, trained using preference data collected from both the LLM and the retriever. Gasket serves as an information bottleneck to control the behavior of both the retriever and the LLM, aligning them with the ultimate goal—generating accurate answers. The Gasket model selects useful context from the passages returned by the retriever, using this context to enhance the original query. The retriever then performs a second round of retrieval based on the enhanced query. Gasket subsequently filter the context, which is finally passed to the LLM to generate the final answer. We designed a method for collecting high quality preference data that allows the Gasket model to be trained offline. We train the Gasket model using a weighted Kahneman-Tversky optimization (KTO) (Ethayarajh et al., 2024). This new practice significantly enhances the stability and data-efficiency of our approach.

In summary, our contributions are as follows:

- We propose a novel method, GasketRAG, which uses an intermediate model to control the data flow in the RAG pipeline, taking into account the preferences of both the LLM and the retriever, thereby improving their collaborative performance in generating answers.

- Our preference collection method ensures high data quality and training efficiency, avoiding the complexity and instability of joint training.

- We meticulously designed experiments to conduct a fair comparison between GasketRAG and the latest RAG methods.

## 2 RELATED WORK

Existing RAG optimization methods can be categorized into three main types: retriever optimization, refinement, and adaptive RAG.

**Retriever Optimization**  D2LLMs (Liao et al., 2024) combine the efficiency of bi-encoders with the nuanced understanding of LLMs in semantic search by decomposing and distilling an LLM cross-encoder into a bi-encoder. Search-Adaptor (Yoon et al., 2024a) customizes the embeddings generated by LLMs for retrieval tasks. Landmark Embedding (Luo et al., 2024) introduces a three-stage method to train LLaMA-2 (Touvron et al., 2023), enabling it to embed sentences within a window of context for retrieval purposes. ARL2 (Zhang et al., 2024a) aligns retrievers with LLMs by leveraging LLM-labeled relevance training data. Additionally, Zhang et al. (2024b) train a multi-task embedder using a rank-aware reward that incorporates LLM feedback. However, all of these methods require retraining the retriever, which is computationally expensive. In contrast, our approach can utilize any off-the-shelf retriever, eliminating the need for retriever retraining and significantly reducing computational costs.

Beyond retriever training, some studies focus on rewriting queries to enhance retrieval quality. For instance, CONQRR (Wu et al., 2022) and RRR (Ma et al.) apply reinforcement learning to optimize query rewriting models based on feedback. Without requiring additional training, LLM4CS (Mao et al., 2023) prompts the LLM to generate multiple query rewritings and synthesizes their embeddings as input for the retriever. While query rewriting can help address the alignment issue between the retriever and LLM, it falls short in performing fine-grained filtering of retrieval results, leaving noise in the retrieved documents unaddressed.

**Refinement**  Refinement involves extracting useful information from lengthy retrieved documents. Zhu et al. (2024a) propose filtering noise by minimizing the mutual information between the refined text and the retrieved passages, while maximizing the mutual information between the refined text and the true answer. Similarly, CFIC (Qian et al., 2024) trains a refiner by selecting sentences based on the generation probabilities of prefix tokens. BGM (Ke et al., 2024) selects documents from the retrieved set and aligns them with downstream task metrics. Beyond refinement, Info-RAG (Xu et al., 2024c) integrates the knowledge from the retrieved passages with the LLM's parameters to improve its robustness when dealing with noisy retrieval results. Fang et al. (2024) also aim to enhance LLM robustness by constructing a noisy dataset and using adversarial training. However, these methods focus exclusively on making the LLM a better document reader, while overlooking the retrieval preferences of the upstream retriever.

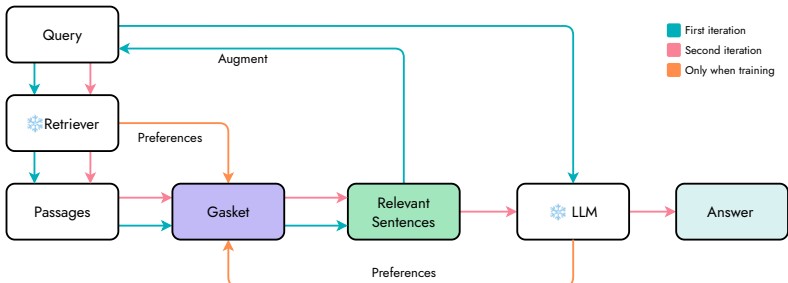

Figure 1: GasketRAG pipeline. GasketRAG involves two iterations. In the first iteration of the outer loop (green line), the gasket model performs an initial filtering of the retrieval results, selecting background information that is beneficial for answering the question and enhancing the retrieval results. This is followed by the second iteration of the inner loop (red line), where the enhanced query is used to re-retrieve documents, which are then filtered by the gasket model again before the LLM generates the answer. The orange line represents the collection of preference data from the outputs of the LLM and Retriever, which is used to train the gasket model.

**Adaptive RAG**  This approach often involves constructing search paths or iteratively interacting between the retriever and the LLM in an adaptive manner. Self-RAG (Asai et al., 2023) introduces special tokens that enable the LLM to automatically express its retrieval needs and assess the relevance of retrieved documents. ActiveRAG (Jiang et al., 2023), on the other hand, prompts the LLM to generate a pseudo-sentence, analyzes low-frequency tokens within it, and conducts targeted retrievals to correct factual inaccuracies. Iter-RetGen (Shao et al., 2023) and InteR (Feng et al., 2024) iteratively refine the query using content generated by the LLM, performing additional retrievals until a final answer is produced. Self-Ask (Press et al., 2023) and GenGround Shi et al. (2024) decompose complex questions into simpler sub-questions, repeating the RAG process until a final answer is reached. However, these methods often require numerous iterations and tend to overlook alignment between the retriever and LLM during each RAG operation. In contrast, our approach focuses on optimizing every minimal unit within the RAG pipeline, ensuring better coordination and performance throughout the entire process.

## 3 METHOD

### 3.1 PRELIMINARY

**Preference Training**  Aligning a model $\pi_\theta$ with the preferences is to learn the value function $v(r_\theta(q, y) - \mathbb{E}_Q[r_\theta(q, y')])$, where $r_\theta = \log \frac{\pi_\theta(y|q)}{\pi_{\text{ref}}(y|q)}$ is a implicit reward function, $q$ is the query, $y$ is the response and $\pi_{\text{ref}}$ is the reference model (Ethayarajh et al., 2024). $\mathbb{E}_Q$ represents reference point and $Q(Y|q)$ is a reference point distribution. The loss then is defined as

$$\mathcal{L}(\pi_\theta, \pi_{\text{ref}}) = \mathbb{E}_{q,y \sim D}[a_{q,y} v(r_\theta(q, y) - \mathbb{E}_Q[r_\theta(q, y')])], \tag{1}$$

where $a_{q,y} \in \{-1, +1\}$ is the preferences and $y'$ is the other possible generations. Inspired by this approach, we use it as the LLM-aware (or retriever-aware) loss to align the components in the RAG pipeline.

### 3.2 OVERVIEW

GasketRAG improves the synergy between the retriever and the LLM generator used for answer generation. It trains a key sentence selector to achieve the purpose of the data-flow control, called the gasket model. This helps the retriever accurately find useful passages and provides the LLM-adapted context, enabling the LLM to generate correct answers. We first collect preference data from the LLM and retriever, then train the gasket model offline. Finally, the trained model is integrated into the pipeline to work collaboratively with the LLM and retriever.

Figure 1 depicts how GasketRAG works. A gasket model $G$ is inserted into the RAG pipeline. Given a query $q$, the retriever $R$ returns top-$k$ passages. Then, all the sentences in the passages are assigned

a unique sentence IDs (SIDs). The top-$k$ passages are rewritten as $Top\text{-}k = \{SID_1 \oplus S_1, SID_2 \oplus S_2, ..., SID_n \oplus S_n\}$, $\oplus$ denotes the concatenation operation. The gasket model will select the sentence IDs related to the query from the top-k passages: $G(q, Top\text{-}k) = \{SID_1, SID_2, ..., SID_m\}$. Subsequently, the query is augmented by the selected sentences: $q' = q \oplus G(q, Top\text{-}k)$ and triggers the second iteration of retrieval and deliver $G(q', Top\text{-}k')$, where $Top\text{-}k' = R(q')$. The newly retrieved passages are processed following the aforementioned method to instruct the gasket model re-select relevant sentences. Finally, the selected sentences are input to the LLM to generate an answer $Answer = LLM(q, G(q', Top\text{-}k'))$. Note here we replace the sentence IDs with the referred sentences.

### 3.3 PREFERENCE COLLECTION

Preference learning refers to the task of predicting an order relation over a collection of objects. In the RAG pipeline, learning the LLM's preference involves understanding how to craft prompts that guide the LLM to generate the desired answer. Learning the retriever's preference focuses on determining how to enhance queries so that the retriever produces high-quality retrieval results. The desired answer is well-defined because we can easily obtain the ground truth. However, the retriever's output usually lacks explicit labels. Labels for relevant documents are scarce and annotated by human. Using LLMs to directly annotate relevant documents is computationally expensive and carries a risk of bias since it is not aligned with the final objective. Therefore, we use an approach to implicitly collect retriever preferences and directly align them with the output objectives of the LLM.

**LLM preference**   Given the query $q$ and the selected relevant sentences $y = \{s_1, s_2, ..., s_m\}$, the selection will be labeled as preferred if the LLM answers the query correctly otherwise dispreferred. Once an LLM preference dataset $D_l = \{(q_i, y_i, a_i)\}|_{i=0}^n$, where $a_i \in \{-1, +1\}$, is collected, the gasket model learns a policy $\pi_\theta$ to minimize the loss (Eq. 1).

**Retriever preference**   Similarly, the sentence selection will be considered as preferred if the augmented query leads to better results in the second retrieval. However, golden passages are not always easily obtainable and often rely on manual annotation, which can result in significant workload and cost. Therefore, we indirectly collect the retriever's preferences by comparing the answers generated by the LLM using the results from the two retrieval iterations and analyzing the distribution of useful information within the retrieved results. The details will be explained later. Similarly, the retriever preference dataset is denoted by $D_l = \{(q_i, y_i, a_i)\}|_{i=0}^n, a_i \in \{-1, +1\}$.

However, directly synthesize the two datasets $D_l$ and $D_r$ to train the gasket model would introduce noise into the preferences of the LLM and retriever. Therefore, we extended the preference label $a \in \{strong\ preferred,\ weak\ preferred,\ weak\ dispreferred,\ strong\ dispreferred\}$, changing it from binary to a four-value system. If both the LLM and retriever prefer the gasket selection result, it's labeled as strong preferred. If only the LLM prefers it, it's labeled as weak preferred. If only the retriever prefers it, it's labeled as weak dispreferred. If neither prefer it, it's labeled as strong dispreferred. Thus, we combine $D_l$ and $D_r$ to create a unified dataset $D$, which integrates the preferences of both the LLM and the retriever.

Figure 2 presents the workflow of preference data collection. Given a query $q$, we first sample multiple different sentence lists $Y = \{y_i | y_i \sim G(q, Top\text{-}k)\}|_{i=0}^n$ from the gasket. The LLM then generates answers based on these sentence lists, which are used to preliminarily label them as preferred or dispreferred. Next, we assign weights (strong or weak) to these labeled sentence lists. Similar to the inference process, each sentence list enters a second iteration. These sentence lists are added to the original query, forming a corresponding number of augmented queries $\{q_i'\}|_{i=0}^n$. The retriever uses the augmented queries to retrieve relevant passages, and the gasket generates new sentence lists $Y' = \{y_i'\}|_{i=0}^n$. Note that for each augmented query, only one sentence list is generated. Next, re-generated sentence lists $Y'$ will be divided into two groups based on the corresponding label of $y$ (preferred or dispreferred). In the preferred group, for each $y'$, we calculate the average sentence index number. A lower average index number indicates that the retriever ranks useful information higher, thus reflecting better retrieval quality. The first and last $y'$ in the sorted group are selected as strong preferred and weak preferred, respectively.

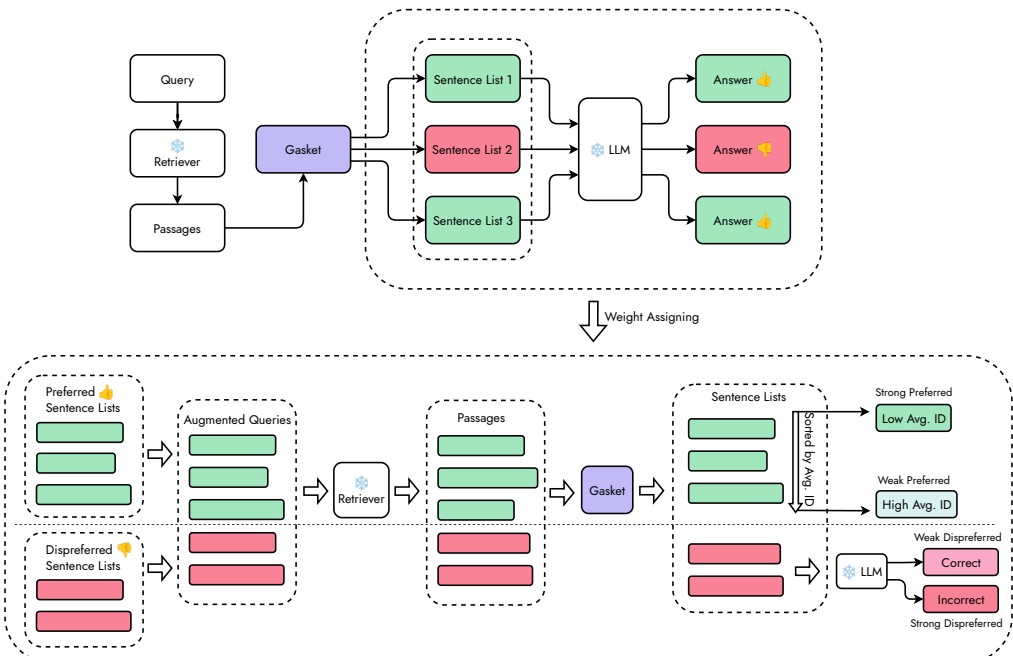

Figure 2: Preference collection. The collection process is divided into two steps, corresponding to two iterations of inference. In the upper part of the diagram, the selection of the gasket is labeled as either "preferred" or "dispreferred." In the lower part, weights are assigned to the samples of these two labeled groups.

For the dispreferred group, the selection is simpler. We ask the LLM to answer the original query $q$ again based on $y'$. If the answer is correct this time, it indicates that the gasket's filtering has effectively improved retrieval quality, and the corresponding $y'$ is labeled as weak preferred. If the LLM still produces an incorrect answer, $y'$ is labeled as strong dispreferred.

## 3.4 GASKET OPTIMIZATION

The gasket model is tasked with two optimization objectives: providing knowledge aligned with the preferences of the LLM, and offering query background aligned with the preferences of the retriever. We focus the tasks of achieving these two objectives within a single model, rather than using two models for joint training, which greatly simplifies the overall pipeline and reduces the difficulty of optimization. Sentence IDs serve as an information bottleneck, restricting the action space of the gasket, enabling it to effectively control the behavior of both the LLM and the retriever, and making it easier for the gasket itself to be optimized from preference data.

**Weighted Kahneman-Tversky Optimization** We use the Kahneman-Tversky Optimization (KTO), which directly maximizes the utility of generations instead of the log-likelihood of preferences, to optimize the model. This also simplifies the framework's complexity and improves data utilization efficiency. Recalling the loss in Eq. 1, KTO uses a biased method to estimate the expectation $\mathbb{E}_Q$ of the reward function $r_\theta$. The estimation is defined as

$$\hat{z}_0 = \max \left( 0, \frac{1}{m} \sum_{i \neq j} \log \frac{\pi_\theta(y_j|q_i)}{\pi_{\text{ref}}(y_j|q_i)} \right),$$

where $q_i$ and $y_j$ are mismatched within a batch of samples. When the distribution of $p(y_i|q_i)$ within a batch shows significant variation, these mismatched pairs $(y_j, q_i)$ will result in both $\pi_\theta(y_j|q_i)$ and $\pi_{\text{ref}}(y_j|q_i)$ having very small values. As a result, the bias in the estimate of $z_0$ will be significantly large. However, since the gasket model only outputs sentence IDs, the action space is very limited,

and the distribution of $p(y_j|q_i)$ does not vary significantly, which naturally addresses this issue and allows for an relatively accurate estimation of $\mathbb{E}_Q$. Originally, KTO accepted binary preference data. To accommodate this, we weighted the training loss for the gasket model:

$$\mathcal{L}_G(\pi_\theta, \pi_{\text{ref}}) = \mathbb{E}_{q,y\sim D}[w \cdot a_{q,y} v(r_\theta(q, y) - \hat{z}_0], \tag{2}$$

where $a_{q,y} \in \{-1, +1\}$ is the preference and $w \in \{0.5 \text{ (weak)}, 1 \text{ (strong)}\}$.

## 4 EXPERIMENTS

We design experiments to compare the performance of GasketRAG with other RAG methods.

### 4.1 SETUP

**Datasets** We perform evaluations on six datasets across three tasks: 1) OpenQA: **TriviaQA** (Joshi et al., 2017) and **PopQA** (Mallen et al., 2023); 2) Multi-hop QA: **HotpotQA** (Yang et al., 2018) and **WikiMultiHopQA** (Ho et al., 2020); 3) Fact-checking: **PubHealth** (Kotonya & Toni, 2020) and **StrategyQA** (Geva et al., 2021). We only used the training data from TriviaQA and HotpotQA to optimize the gasket model, while other test datasets are treated as independent evaluations. Due to the time-consuming testing process, we only use the first 1000 queries from each dataset.

**Preference Data** We used the training sets of TriviaQA (Joshi et al., 2017) and HotpotQA (Yang et al., 2018) to collect preference data, utilizing only the queries and answers from these datasets. LLaMA-3.1-8B-Instruct was employed as the gasket model to generate sentence selections. We follow Zhang et al. (2024c) to finetune LLaMA-3-8B as the answer generator. Since the LLM sometimes does not follow the exact standard answer format, to avoid labeling correct answers as dispreferred due to differences in expression or output format, we used not only Exact Match for evaluation but also leveraged ChatGPT to assess the correctness of the answers. For each query, we generate 5 samples. If all the answers are correct, we consider the query too simple and not requiring optimization, so it is discarded. Similarly, if all 5 samples are labeled as dispreferred and the answers remain incorrect after the second iteration, we consider the query too difficult to effectively train the gasket model and discard it as well. Table 1 presents the statistics of the collected preference data. The difficulty is determined by how many incorrect answers are found among the five sentence selection samples, presented as a percentage. During the data collection process, we observed that TriviaQA generated more samples with entirely correct answers (which were discarded) compared to HotpotQA, indicating that TriviaQA is easier than HotpotQA. Therefore, considering data collection efficiency and balancing difficulty, we collected only 5k samples from TriviaQA and 12k samples from HotpotQA.

| | Query Source | | |
| --- | --- | --- | --- |
| | TriviaQA | HotpotQA | Overall |
| Samples | 5006 | 12805 | 17811 |
| Strong / Weak Preferred Samples | 1396 / 947 | 3466 / 2278 | 4862 / 3225 |
| Strong / Weak Dispreferred Samples | 1290 / 1373 | 3353 / 3708 | 4643 / 5081 |
| Avg. Difficulty | 52.19 | 54.61 | 53.93 |
| Avg. Prompt Length | 1173.67 | 1177.74 | 1176.59 |
| Avg. Sentences | 63.68 | 65.36 | 64.88 |
| Avg. Selected Sentences | 3.65 | 3.81 | 3.76 |

Table 1: Preference data statistics.

**Metrics** We use Accuracy (**ACC**) as the metric to evaluate the model's responses, which is determined by checking whether the correct answer is included in the model's generated content. However, due to the variability in the model's responses, the Accuracy metric may introduce errors. Similar to the strategy used when collecting preference data, we additionally use ChatGPT to assess whether the model's response aligns with the reference answer, thereby calculating the **Correctness** score.

## 4.2 BASELINES

We compare GasketRAG with several baselines: 1) **Direct**, where we prompt the LLM to answer the question without retrieval; 2) **NaiveRAG**, a standard RAG process, where the retrieved passages are passed to the LLM without augmentation; 3) Rewrite-Retrieve-Read (**RRR**) (Ma et al.), a method for aligning the retriever and LLM through query rewriting.; 4) **Iter-RetGen** (Shao et al., 2023), which synergizes retrieval and generation in iterations; 5) **ActiveRAG** (Jiang et al., 2023), a method that predicts the next sentence to anticipate future content and uses it as a query to retrieve documents, regenerating the sentence if it contains low-confidence tokens; 6) **SelfAsk** (Press et al., 2023), which improves chain-of-thought, where the model proposes follow-up sub-questions to solve before arriving at the final answer. 7) **SelfRAG** (Asai et al., 2023), enhancing RAG performance through adaptive retrieval and self-reflection; 8) **SearChain** (Xu et al., 2024b), which builds a reasoning chain to iteratively propose unsolved sub-questions and verify the answer with the retrieval information.

SelfRAG requires an LLM generator with additional special tokens, meaning the model undergoes extra finetuning. To ensure a fair comparison, we follow Zhang et al. (2024c) to finetune unified generators using the same training data as Asai et al. (2023). We also use ChatGPT as a generator to evaluate the different methods. SearChain was reproduced using ChatGPT as the backend.

**Implementation Details**   The gasket model training runs on a 4-GPU H100 node. The base model is *LLaMA-3.1-8B-Instruct*[1]. For sufficient KL-divergence estimate in a KTO step, the batch size is set to 2 per GPU with 4 gradient accumulation steps. Low-Rank Adaptation (LoRA) is utilized where the rank and the scaling factor are 16, targeting all linear layers. The learning rate is $1e - 5$, with a warmup ratio of $0.1$. For preference data collection, *GPT-3.5-turbo* serves as the gasket model to generate sentence selections and as the discriminator to evaluate the correctness of the responses. ColBERTv2 (Santhanam et al., 2022) is employed as the retriever. We use the 2018 Wikipedia corpus provided by Karpukhin et al. (2020), where the documents are chunked into passages with a maximum length of 100 words. For all methods we use the top-10 retrieved passages.

## 4.3 RESULTS

| Method | PopQA† | | TriviaQA | | HotpotQA | | WikiMultiHop† | | PubHealth† | StrategyQA† |
|---|---|---|---|---|---|---|---|---|---|---|
| | ACC | Correctness | ACC | Correctness | ACC | Correctness | ACC | Correctness | ACC | ACC |
| *w/ LLaMA-3-8B* | | | | | | | | | | |
| Direct | 29.2 | 34.1 | 66.7 | 61.6 | 20.1 | 40.6 | **25.0** | 37.3 | **73.2** | 55.7 |
| NaiveRag | 36.6 | 43.7 | 63.2 | 61.0 | 26.7 | 47.5 | 20.1 | 32.0 | 67.8 | 57.8 |
| RRR | 35.9 | 43.0 | 58.8 | 56.0 | 23.2 | 42.8 | 21.9 | 29.8 | 68.3 | **58.2** |
| Iter-RetGen | 34.2 | 43.5 | 64.9 | 61.0 | 27.0 | 47.0 | 20.6 | 32.9 | 41.8 | 55.8 |
| ActiveRAG | 35.2 | 44.6 | 64.1 | 60.9 | 26.5 | 47.3 | 18.7 | 30.1 | 47.6 | 56.2 |
| SelfAsk | 11.2 | 16.6 | 36.0 | 36.1 | 14.6 | 30.2 | 17.6 | 26.9 | 40.6 | 52.1 |
| SelfRAG | 34.9 | 37.4 | 56.0 | 50.8 | 21.8 | 38.4 | 20.4 | 22.1 | 64.9 | 46.7 |
| GasketRAG (ours) | **39.1** | **45.7** | **67.9** | **65.5** | 29.8 | **54.8** | 22.7 | **38.6** | 72.1 | 58.1 |
| *w/ GPT-3.5-turbo* | | | | | | | | | | |
| Direct | 32.0 | 37.0 | **77.5** | **71.7** | 31.8 | 54.4 | 37.0 | 42.7 | **77.7** | **68.0** |
| NaiveRag | 44.0 | 45.6 | 72.6 | 66.5 | 38.4 | 58.9 | 32.7 | 37.5 | 53.9 | 61.2 |
| RRR | 44.7 | 46.2 | 71.9 | 65.9 | 38.0 | 58.4 | 31.2 | 36.5 | 53.7 | 63.3 |
| Iter-RetGen | 43.7 | 45.1 | 73.5 | 67.5 | **43.7** | 61.0 | 35.8 | 38.5 | 42.6 | 56.9 |
| ActiveRag | 43.7 | 45.1 | 73.6 | 67.8 | 42.8 | 61.0 | 35.0 | 39.2 | 50.4 | 62.0 |
| SelfAsk | 35.9 | 41.4 | 66.3 | 62.0 | 36.7 | 57.2 | **39.1** | 42.8 | 46.3 | 38.4 |
| SearChain | 31.7 | 43.7 | 66.3 | 63.9 | 33.5 | 59.2 | 32.0 | 44.5 | 30.1 | 60.5 |
| GasketRAG (ours) | **44.8** | **48.2** | 73.8 | 69.3 | 42.8 | **62.8** | 37.3 | **48.0** | 64.3 | 61.9 |

Table 2: Performance comparison between GasketRAG and various RAG methods. † indicates we do not use the training sets of those datasets to optimize the gasket model. The best scores are highlighted in bold, while the second-best scores are underlined.

Table 2 gives the results of performance evaluation of GasketRAG and the baselines. Our findings are as follows.

**Effectiveness of GasketRAG**   It can be observed that our approach outperforms the previous RAG methods on most test sets and metrics. Apart from the TriviaQA and HotpotQA datasets, where the

---

[1]https://huggingface.co/meta-llama/Meta-Llama-3.1-8B-Instruct

gasket model was trained using their training sets, significant improvements were also observed on other test sets. Moreover, GasketRAG demonstrated high stability when handling different types of questions.

**Strong Direct and NaiveRAG** We observed that both Direct and NaiveRAG exhibited strong performance across multiple test sets. We believe this is primarily because the LLM's parameters already contain knowledge relevant to these datasets. Additionally, since we used the top-10 retrieved passages, the input became lengthy and included a lot of irrelevant information. As a result, when using RAG methods, the LLM may be distracted. Furthermore, when provided with retrieved context, the LLM tends to suppress its parametric knowledge (Tan et al., 2024), leading to a decrease in performance. Another reason could be that the 2018 Wikipedia corpus lacks relevant documents. For instance, PubHealth involves a substantial amount of biomedical knowledge, which the retriever may struggle to provide effectively. Nevertheless, GasketRAG still shows significant advantages over other methods. On PopQA (ACC), TriviaQA (ACC and Correctness), HotpotQA (Correctness) and WikiMultiHop (Correctness), it is the only method (with LLama-3 as the generator) that surpasses both Direct and NaiveRAG.

**Generalization** Although the gasket model was trained based on the preferences of LLaMA-3-8B, it still demonstrates good generalization performance when the generator is replaced with GPT-3.5-turbo. This is because, during preference learning, the gasket model develops a stronger ability to filter out irrelevant information, which is a key factor in improving the performance of the RAG pipeline. As a result, a gasket model trained on the preferences of one LLM can still provide benefits to other LLMs.

## 4.4 EFFECTIVENESS OF PREFERENCE TRAINING

We compare the impact of the gasket model trained with KTO and its base model (LLaMA-3.1) on the performance of GasketRAG. We also used GPT-3.5 as a gasket model for evaluation. Table 3 presents the result. It can be observed that preference alignment significantly improves the performance of LLaMA-3.1. Without training, LLaMA-3.1 underperforms NaiveRAG across all tasks. It can be observed that using a stronger model (GPT-3.5-turbo) did not result in a substantial performance improvement. This highlights the importance of eliminating the preference gap between the retriever and the LLM through preference learning. Furthermore, this underscores that GasketRAG is not merely a simple refinement or rewriting approach.

| Model | PopQA | | TriviaQA | | HotpotQA | | WikiMultiHop | | PubHealth | StrategyQA |
|---|---|---|---|---|---|---|---|---|---|---|
| | ACC | Correctness | ACC | Correctness | ACC | Correctness | ACC | Correctness | ACC | ACC |
| GPT-3.5-turbo | 33.9 | 42.1 | 66.5 | 63.3 | 27.4 | 49.4 | 19.9 | 35.0 | 69.6 | 57.5 |
| LLaMA-3.1-8B-Instruct | 35.6 | 42.3 | 63.6 | 60.9 | 24.3 | 47.5 | 24.9 | 37.8 | 69.6 | 57.4 |
| Gasket | 39.1 | 45.7 | 67.9 | 65.5 | 29.8 | 54.8 | 22.7 | 38.6 | 72.1 | 58.1 |

Table 3: Performance comparison between the gasket model and the base model.

## 4.5 EFFECTIVENESS OF WEIGHTED KTO

We retrained the gasket model with the exact same settings, but ignore the sample weights during loss calculation. Figure 3 exhibits the difference between weighted and non-weighted KTO methods. The weighted KTO training demonstrates a clear advantage over the non-weighted version. By further distinguishing between weak and strong preferences in the binary preference data, the distraction of weak preference samples is reduced, allowing the gasket model to converge more effectively. Additionally, the non-weighted trained gasket model still shows improvements over NaiveRAG.

We also train a gasket model with SFT. The result (Figure 3) demonstrates a significant performance degradation with SFT. SFT provides an unbiased estimation of the target preference, capturing general trends in labeled data. However, it offers a biased estimation for the model, as it does not account for the model's inherent limitations or specific dynamics. This limitation means that SFT cannot enable the model to precisely adapt to subtle differences in preferences.

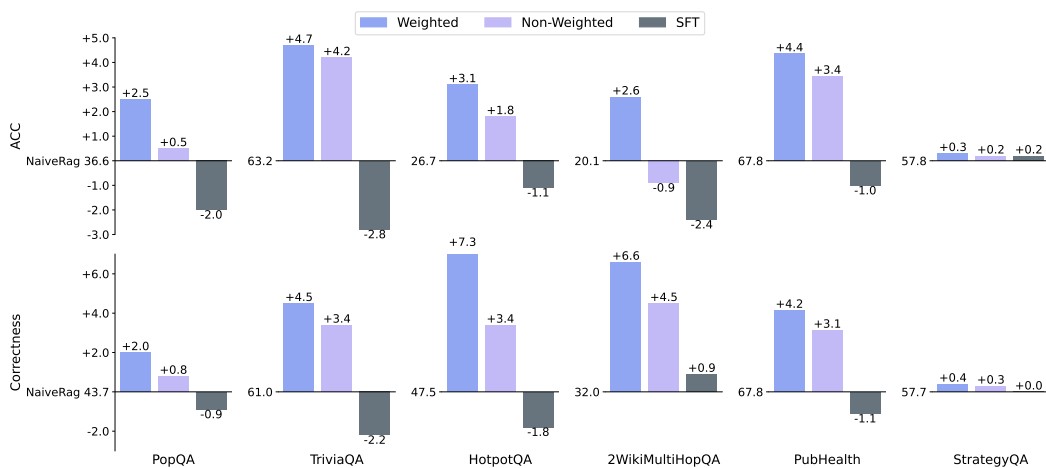

Figure 3: Comparison between weighted KTO, non-weighted KTO and SFT trained GasketRAG.

## 4.6 MODEL AND TRAINING DATA SIZE SCALING

We additionally trained two gasket models based on Qwen-2.5: a 0.5B and a 1.5B model. These were compared with our gasket model based on LlaMA-3.1. From Figure 4a, it can be observed that as the model size increases, the performance also shows improvement. The 0.5B model has performance limitations, indicating that regulating the retriever and LLM within the RAG pipeline requires the gasket model to possess a considerable level of text understanding. Additionally, it is evident that the 1.5B model demonstrates quite strong performance, slightly lagging behind the 8B model across datasets but still outperforming other RAG methods. This indicates that GasketRAG is more efficient compared to other RAG methods. By using a smaller model as the gasket model to filter sentences, the token sequence length input to the LLM generator is significantly reduced. As a result, despite requiring two iterations, GasketRAG has light resource demands.

To study the impact of training data sizes on the gasket model. We retrained the gasket model based on LLaMA-3.1-8B with half the preference data (8.5K). The results are shown in Figure 4b. The performance gap between the gasket model trained with 8.5K data and the one trained with 17K data is minimal, demonstrating the high data efficiency of our proposed preference data collection method and the Weighted KTO training algorithm. GasketRAG requires only a small amount of preference data to achieve a high level of performance.

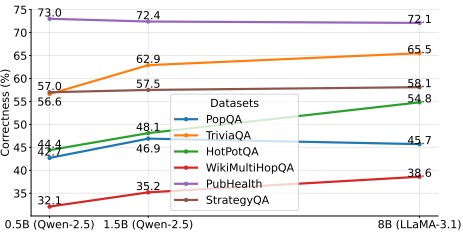

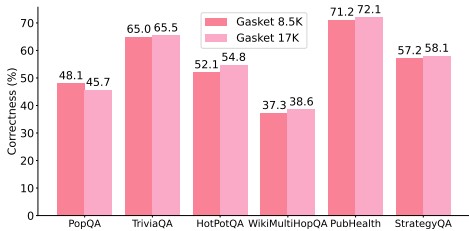

(a) Percentage change of different model sizes relative to 0.5B model

(b) Gasket models on different training dataset sizes.

Figure 4: Correctness comparisons between different model and training data sizes.

## 4.7 EFFECTIVE OF DIFFERENT ITERATIONS

Table 4 reveals the performance of GasketRAG when different number of iterations are applied. Only one iteration means there is no query augmentation and a second retrieval, where the gasket model only functions as a information filter. It can be observed that the 2-Iteration GasketRAG achieves the best overall performance. Information that was not accurately retrieved in the first

iteration is often identified after an additional round of adjustment. However, increasing the number of iterations can also lead to the accumulation and amplification of errors, resulting in some performance degradation.

| Iterations | HotpotQA | | WikiMultiHopQA | | PopQA | | TriviaQA | |
|---|---|---|---|---|---|---|---|---|
| | ACC | Correctness | ACC | Correctness | ACC | Correctness | ACC | Correctness |
| 1 | 28.4 | 52.9 | 19.1 | 34.7 | 38.7 | 46.4 | 68.3 | 65.7 |
| 2 | 29.8 | 54.8 | 22.7 | 38.6 | 39.1 | 45.7 | 67.9 | 65.5 |
| 3 | 29.6 | 52.6 | 21.2 | 36.2 | 37.1 | 44.8 | 68.4 | 64.3 |

Table 4: Iterations of GasketRAG comparison.

## 4.8 LATENCY ANALYSIS

We measured the latency of each RAG method, as shown in Figure 5. We ran all tests on a single H100 GPU and recorded the average time the method took from issuing a query to receiving the final answer. It is worth noting that we used the vLLM (Pisarchyk & Lee, 2020) engine's API server with 20 concurrent threads, so the times shown in the graph include the waiting time for requests in the queue. In the case of synchronous inference, a typical 2-Iteration GasketRAG processes a query in approximately 0.8 seconds. It can be observed that GasketRAG has slightly higher latency compared to SelfAsk and Iter-RetGen. However, the 1-Iteration Gasket is significantly faster than both while also delivering better performance.

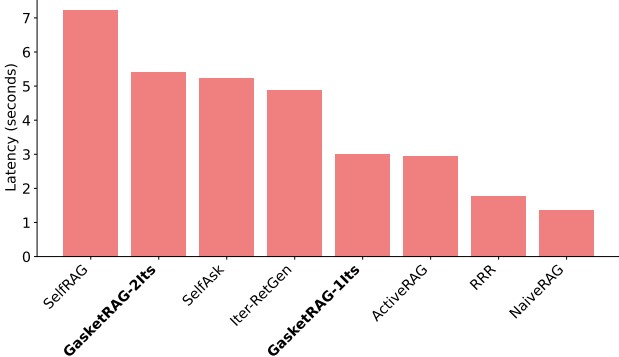

Figure 5: The average processing latency of each sample.

## 5 CONCLUSION

In this paper, we propose a new method, GasketRAG, which systematically aligns all components of the RAG pipeline in an end-to-end manner. By collecting preference data between the LLM and retriever, we perform Weighted KTO training to obtain a gasket model that effectively coordinates the RAG process. Our approach eliminates the need for complex joint training and costly data annotation. Through rigorous and fair comparisons in our experiments, the results show that GasketRAG significantly outperforms other methods. By comparing with strong LLMs without preference training, we demonstrate the importance of aligning LLMs with retrievers to address the preference gap. We trained gasket models based on LLMs of various parameter scales, showing that even a much smaller gasket model can achieve performance surpassing other RAG methods. Additionally, we reveal the high data efficiency of GasketRAG, achieving training objectives with only a small amount of preference data. Due to resource limitations, we have not yet explored GasketRAG's performance on more complex and domain-specific tasks, which will be left for future work.

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
