# OpenReview forum: "GasketRAG: Systematic Alignment of Large Language Models with Retrievers"
_ICLR.cc/2025/Conference — ICLR 2025 Conference Withdrawn Submission_

### Official Review · Reviewer_UHMc · 2024-10-28

**Soundness:** 2
**Presentation:** 2
**Contribution:** 2
**Rating:** 5
**Confidence:** 3

**Summary:**

The IR-LLM mismatch is a critical problem for RAG systems. In this paper, the authors propose to introduce a gasket model between retrievers and LLMs to enhance the RAG performance. Specifically, the Gasket model learns to select key sentences preferred by both LLMs and retrievers and iteratively generate context-augmented queries, and a data construction algorithm and a learning algorithm are designed to train the Gasket model. Experiments show some performance improvements over baselines.

I raise my score from 3 to 5 due to the rebuttal, but  my main concerns are still here for the over-claim problem and the incremental contribution.

**Strengths:**

In general, I think it is important to address the IR-LLM mismatch problem, and the idea of leveraging a gasket model is reasonable just like the classical copilot paradigm.  Furthermore, I think the learning algorithm and the data construction method for learning set preference with hidden labels are also helpful.

**Weaknesses:**

1. The idea of iteratively refining/rewriting queries is not new, although previous studies may not train their refining/rewriting models;
2. Relevant context selection/identification is also not a new idea. There are many related studies in long context LLMs and RAG methods;
2. Although the title is "systematic alignment of LLMs with retrievers", I don't think LLMs and Retrievers can be systematically aligned by only rewriting queries and selecting sentences;
3. The performance improvement in experimental results is not significant, especially comparing with naive RAG. The authors should analyze these surprising results in detail.
3. Furthermore, I think the baselines are too weak and the results are not convincing, the authors should at least compare their methods with: 1) the long context models which can identify and leverage information in long contexts; 2) RAG methods with denoising and iterative query rewriting components; 3) the same model of GasketRAG, but replacing the Gasket model with a strong LLM (e.g., GPT 3.5, Llama 3, etc.) without any training.

**Questions:**

See Weaknesses in above.

---

> ### Author Response · Authors · 2024-11-16
>
> Thank you for your thoughtful review and feedback on our submission. We appreciate the time and effort you invested in understanding our work. We would like to address the concerns you raised:
>
> **Concern 1: Novelty of our method.**
>
> **Response:**
>
> We would like to address potential misunderstandings about our work and provide clarity on the unique contributions of GasketRAG.
>
> First and foremost, GasketRAG is **not merely a combination of refining and rewriting**, and we have never claimed that our primary contribution lies in proposing context selection or identification techniques. Instead, our research focuses on a critical challenge: **bridging the preference gap between retrievers and large language models (LLMs)** and enhancing the performance of off-the-shelf retrievers in collaboration with LLMs.
>
> We understand the perception that the Gasket model is primarily about selecting relevant sentences. However, the crux of the issue lies in how "relevance" is defined and how to fine the optimal “relevance”. Retrievers are typically optimized based on human preferences and are designed to retrieve and rank documents in ways that align with human habits. However, **the preferences of LLMs differ from those of humans**, leading to a misalignment.
>
> The Gasket model addresses this misalignment not through simple selection or rewriting but by **manipulating the content and noise** in inputs to both the retriever and the LLM. This manipulation modulates their behavior, effectively bridging the preference gap between the two systems.
>
> Traditional approaches to addressing this challenge often involve **jointly training the retriever and LLM** using techniques such as PPO and reward models. While effective, these methods significantly increase the complexity and instability of the framework, creating barriers to research and scalability in this area.
>
> Our innovation lies in taking a simpler yet highly effective approach:
>
> 1. **No reliance on human annotations**, which may not adequately address the preference gap.
> 2. **No need for complex joint training frameworks** involving all components of the RAG pipeline.
> 3. Introduction of a **straightforward and efficient method for collecting preference data**.
> 4. Utilization of a novel **Weighted KTO method** to optimize the interaction between the retriever and LLM.
>
> Our experimental results demonstrate that GasketRAG outperforms traditional iterative retrieval, chain-of-thought (CoT) reasoning, and question rewriting methods. Moreover, it offers **greater stability** across different datasets and LLMs, further solidifying its utility in real-world scenarios.
>
> ---
>
> **Concern 2: I don't think LLMs and Retrievers can be systematically aligned by only rewriting queries and selecting sentences**
>
> **Response:**
>
> Our experimental results demonstrate the significant effectiveness of the GasketRAG method. As shown in Table 2, GasketRAG (LLaMA-3) outperforms other methods by a large margin on the PopQA, TriviaQA, HotpotQA, and WikiMultiHop test sets, and is very close to the best-performing method on the PubHealth and StrategyQA datasets. Moreover, it exhibits consistent performance on untrained OOD (Out-of-Domain) test sets. We observed that, unlike GasketRAG, other RAG methods perform significantly worse than NaiveRAG on some datasets, highlighting the superior **stability** of GasketRAG.
>
> It is worth noting that our gasket model was trained using preference-aligned data specific to LLaMA-3. However, even when the generator LLM was replaced with GPT-3.5, GasketRAG still achieved superior performance on multiple datasets, demonstrating the excellent **generalization capability** of our approach.

---

> ### Author Response · Authors · 2024-11-16
>
> **Concern 3: The performance improvement in experimental results is not significant, especially comparing with naive RAG.**
>
> **Response:**
>
> We need to point out some **important misunderstandings** here. As shown in Table 2, GasketRAG **demonstrates significant improvements over NaiveRAG across all test sets and metrics**, which is something that no other RAG method has achieved.
>
> ---
>
> **Concern 4: Furthermore, I think the baselines are too weak and the results are not convincing, the authors should at least compare their methods with: 1) the long context models which can identify and leverage information in long contexts; 2) RAG methods with denoising and iterative query rewriting components; 3) the same model of GasketRAG, but replacing the Gasket model with a strong LLM (e.g., GPT 3.5, Llama 3, etc.) without any training.**
>
> **Response:**
>
> In fact, the baselines we selected are all state-of-the-art and representative RAG methods, including different types such as question rewriting, iterative RAG, adaptive RAG and CoT-based RAG. Our experiments were conducted under strictly fair settings.
>
> We will address your three suggestions one by one.
>
> For **1)**,  the average passage length in the corpus we used is 100 words. We configured the retriever to return the top-10 passages, which is **far smaller than the maximum context length** of the LLM generators we used (LLaMA-3 and GPT-3.5). Therefore, none of the baseline methods experienced performance degradation due to context length limitations. We understand your interest in exploring the application of GasketRAG in long-context RAG scenarios, and we agree that this is a fascinating and important direction for future research. However, it falls outside the scope of this study. In this paper, we focus on addressing the **preference gap** between retrievers and LLMs, and we propose a **practical and straightforward preference training method** to align the components of the RAG pipeline effectively.
>
> For **2)**, the baselines we selected already include these two types of RAG methods: RRR and Iter-RetGen. Both of these methods serve as very strong baselines. However, as shown in the results, although these methods exhibit strong performance on specific datasets, their generalization and stability are relatively poor due to the lack of preference alignment between the retriever and the LLM in their approaches.
>
> For **3)**, in Table 3, we present a comparison between the gasket model and its base model, **LLaMA-3.1-8B-Instruct**, which has not undergone additional training. It can be observed that our proposed **preference data collection method** and the **Weighted KTO** training approach significantly enhance the model's performance within the GasketRAG framework. Additionally, we also tested GasketRAG using GPT-3.5 as the gasket model, with the results (correctness) shown below:
>
> |  | GPT-3.5-turbo | LLaMA-3.1-8B-Instruct | Gasket |
> | --- | --- | --- | --- |
> | PopQA | 42.1 | 42.3 | 45.7 |
> | TriviaQA | 63.3 | 60.9 | 65.5 |
> | HotPotQA | 49.4 | 47.5 | 54.8 |
> | 2Wiki | 35.0 | 37.8 | 38.6 |
> | PubHealth | 69.6 | 69.6 | 72.1  |
> | StrategyQA | 57.5 | 57.4 | 58.1 |
>
> It can be observed that using a stronger model did not result in a substantial performance improvement. This highlights the importance of eliminating the preference gap between the retriever and the LLM through preference learning. Furthermore, this underscores that **GasketRAG is** **not merely a simple refinement or rewriting** approach.
>
> We hope our clarifications address your concerns and answer your questions. Thank you once again for your valuable feedback.

---

> > ### Comment · Reviewer_UHMc · 2024-11-28
> >
> > Thanks for the rebuttal.
> > I think the response clarifies some of my concerns. But my main concern is not "GasketRAG is merely a simple refinement or rewriting approach", but I think that the claim "jointly training the retriever and LLM using techniques such as PPO and reward models" can align retriever and LLM is too strong. Actually, there are previous studies which train retriever for LLM-based generation.  I can raise my score, but my main concerns are still the over-claim problem and the incremental contribution.

---

> > > ### Author Response · Authors · 2024-11-28
> > >
> > > Thank you for your response. Below is our explanation addressing the concerns you raised:
> > >
> > > Regarding "jointly training the retriever and LLM using techniques such as PPO and reward models," this was mentioned as a potential solution. However, due to the inherent complexity and instability of such joint training, we proposed the GasketRAG method as a more stable and data-efficient alternative.
> > >
> > > As for prior research on training retrievers for LLM-based generation, we have also reviewed such methods in our related work section. These approaches tend to have high data complexity because training the retriever requires a large amount of pairwise data. Our method avoids this requirement. As demonstrated in our experiments (Figure 4b), GasketRAG achieves significant improvements even with just 8.5K data points and shows consistent stability across datasets. Additionally, retraining retrievers using LLM-labeled ranking data does not directly target the quality of LLM responses as the objective.
> > >
> > > We have not overclaimed our contributions. All our claims are substantiated by corresponding experiments. We would like to understand which specific contribution you perceived as overclaimed, so we can address it further.
> > >
> > > Thank you once again for your response and for taking the time to review our paper.

---

> ### Author Response · Authors · 2024-11-26
>
> Dear reviewer,
>
> I hope this message finds you well.
>
> We wanted to kindly follow up to see if you have had a chance to review our response to your comments. We would greatly appreciate any feedback regarding whether we have adequately addressed the concerns you raised in your review.
>
> Your insights are invaluable to us, and if you have any additional questions or suggestions, we would be more than happy to address them.
>
> Thank you once again for your time and thoughtful consideration.
>
> Best regards,
>
> All Authors

---

### Official Review · Reviewer_kEed · 2024-11-01

**Soundness:** 2
**Presentation:** 3
**Contribution:** 3
**Rating:** 6
**Confidence:** 3

**Summary:**

The authors propose a framework which attempts to adjust what sentences are presented to an LLM based on signal about whether those sentences allow the LLM to produce the correct answer or not. GasketRAG fine-tunes an intermediate LLM using preference training (Weighted KTO optimization) to improve the model's ability to choose which sentences are relevant to a specific query from a set of retrieved passages.
This leads to improvements over previous RAG systems on several question answering datasets.

**Strengths:**

- The problem addressed by this paper is interesting and important, it is clear that future RAG system must close the loop between retriever and LLMs. This method introduces an attempt to solve this important issue.

**Weaknesses:**

- The authors claim that the "preference" from the retriever is being used within the data collection but it seems like the only signal which decides between preferred and dis-preferred is whether an LLM can use the retrieved sentences to answer the query correctly. I do not think the Gasket is being trained on any retriever preference signal (please explain if you believe this to be a misconception).
- The GasketRAG concept is not very novel, it is actually a simple re-ranking module which helps determine which retrieved documents are relevant to a specific query. There are many such methods which leverage LLMs such as RankGPT (Sun et al 2023) and they should be used as baselines in this work.
- Preference training might be sophisticated and interesting but the motivation to use it is unclear. It seems to me that simple SFT could also allow for the "gasket" model to identify which textual chunks are relevant for the query.
 - Why is the "Direct" performance of both models in Table 2 sometimes stronger than the NaiveRAG implementation? This is very strange behavior, especially in factual QA datasets like these ones.

**Questions:**

See weaknesses

---

> ### Author Response · Authors · 2024-11-16
>
> Thank you for your thoughtful review and feedback on our submission. We appreciate the time and effort you invested in understanding our work. We would like to address the concerns you raised:
>
> **Concern 1: Preference of the retriever.**
>
> **Response:**
>
> It seems there may have been some misunderstanding of our method. Let us clarify, starting from the definition of **preference learning**. Preference learning refers to the task of predicting an order relation over a collection of objects.
>
> In the RAG pipeline:
>
> - **Learning the LLM's preference** involves understanding how to craft prompts that guide the LLM to generate the desired answer.
> - **Learning the retriever's preference** focuses on determining how to enhance queries so that the retriever produces high-quality retrieval results.
>
> Now, let’s return to GasketRAG. In GasketRAG, the LLM's desired answer is well-defined because we have access to the ground truth. However, the retriever’s output lacks explicit labels. Therefore, what constitutes high-quality retrieval results is indirectly labeled based on whether these results enable the LLM to generate the correct answer. Thus, the preferences of the retriever are implicitly learned. During training, the gasket model learns how to enhance queries by observing the impact of modified input of the retiever on the final result (LLM's answer). This process helps the gasket model guide the retriever to retrieve the desired documents. Therefore, this is a result-oriented approach. We do not focus on how the gasket model improves the retriever; we only reward the gasket model when it performs the correct actions (selected the right sentences).
>
> ---
>
> **Concern 2: Novelty of our method.**
>
> **Response:**
>
> We understand that you believe the gasket model simply selects relevant sentences to help the retriever find more pertinent documents and assist the LLM in generating responses. However, the issue lies in how "relevance" is defined. Retrievers are generally trained based on human preferences, designed to retrieve and rank documents in a way that aligns with human habits. However, the preferences of LLMs do not completely align with those of humans.
>
> As pointed out in [1], when LLMs are presented with both LLM-generated and retrieved documents, they tend to favor content generated by LLMs. This demonstrates subtle differences in how LLMs process contexts from different sources. Furthermore, [2] validates that LLMs are not particularly sensitive to the ranking of documents but are highly sensitive to irrelevant information (noise) within documents. **We define the divergence between the retriever's output preferences and the LLM's input preferences as the *preference gap*.**
>
> Thus, the gasket model does more than merely re-ranking. **Its underlying logic involves manipulating the content and noise in the inputs to both the retriever and the LLM to modulate their behavior and bridge this preference gap.**
>
> Existing methods typically focus on specific components of the RAG pipeline. For instance, some use refinement techniques to help the LLM better understand the content returned by the retriever, while others employ question rewriting to improve the retriever's relevance in retrieving documents. While these approaches have achieved certain improvements, they often overlook the *preference gap* between the LLM and the retriever. Bridging this gap offers additional potential for enhancement.
>
> The traditional solution to this issue involves jointly training the LLM and the retriever using algorithms such as PPO, with reward models incorporated. However, this significantly increases the complexity and instability of the framework, limiting the scope of research in this area.
>
> The innovation of our approach lies in the fact that it does **not rely on human annotations** (which, in fact, may fail to align with the preference gap) **nor on building a complex joint training framework** involving all components of the RAG pipeline. Instead, our method introduces a **straightforward and effective way to collect preference data** and utilizes a **Weighted KTO** method. Our experiments demonstrate that GasketRAG outperforms iterative retrieval, chain-of-thought (CoT) reasoning, and question rewriting methods in terms of performance and exhibits **greater stability** across different datasets and various LLMs.
>
> [1] Hexiang Tan, Fei Sun, Wanli Yang, Yuanzhuo Wang, Qi Cao, and Xueqi Cheng. 2024. Blinded by Generated Contexts: How Language Models Merge Generated and Retrieved Contexts When Knowledge Conflicts?
>
> [2] Zixuan Ke, Weize Kong, Cheng Li, Mingyang Zhang, Qiaozhu Mei, and Michael Bendersky. 2024. Bridging the Preference Gap between Retrievers and LLMs.

---

> ### Author Response · Authors · 2024-11-16
>
> **Concern 3: Simple SFT could also allow for the "gasket" model to identify which textual chunks are relevant for the query.**
>
> **Response:**
>
> Research on SFT and preference training is already well-established [1] [2]. Compared to preference training, SFT represents a more coarse-grained optimization approach. **SFT provides an unbiased estimation of the target preference**, capturing general trends in labeled data. However, **it offers a biased estimation for the model**, as it does not account for the model's inherent limitations or specific dynamics.
>
> This limitation means that SFT cannot enable the model to precisely adapt to subtle differences in preferences. In contrast, preference training, by focusing on fine-grained adjustments through direct feedback, allows the model to better capture nuanced variations, leading to improved alignment with desired behaviors in specific scenarios.
>
> We trained the gasket model using SFT under the same configuration and conducted evaluations. The result (correctness) is as follows:
>
> |  | GasketRAG (Base) | GasketRAG (SFT) | GasketRAG (WKTO) |
> | --- | --- | --- | --- |
> | PopQA | 42.3 | 42.8 | 45.7 |
> | TriviaQA | 60.9 | 58.8 | 65.5 |
> | HotPotQA | 47.5 | 45.7 | 54.8 |
> | 2Wiki | 37.8 | 32.9 | 38.6 |
> | PubHealth | 69.6 | 66.7 | 72.1  |
> | StrategyQA | 57.4 | 57.7 | 58.1 |
>
> As expected, **the gasket model trained with SFT exhibited significant performance degradation.** Compared to the base model (LLaMA-3.1-8B-Instruct), the model trained with SFT performed even worse.
>
> [1] Ethayarajh, K., Xu, W., Muennighoff, N., Jurafsky, D., & Kiela, D. (2024). Kto: Model alignment as prospect theoretic optimization.
>
> [2] Hua, E., Qi, B., Zhang, K., Yu, Y., Ding, N., Lv, X., ... & Zhou, B. (2024). Intuitive Fine-Tuning: Towards Unifying SFT and RLHF into a Single Process.
>
> ---
>
> **Concern 4: Why is the "Direct" performance of both models in Table 2 sometimes stronger than the NaiveRAG implementation?**
>
> **Response:**
>
> We were also quite surprised by this result. However, we observed similar outcomes when testing on multiple models, such as GPT-3.5. Additionally, we noticed that similar results were also reflected in [1]. We hypothesize that the possible reason for this phenomenon is that these LLMs were likely exposed to relevant information during pretraining and may even have encountered the test sets (data leakage). As a result, they can answer some questions correctly even without context. However, when NaiveRAG provides retrieved context, a masking effect occurs [2], causing the LLM to rely more on the contextual information rather than its parameter knowledge. Since the context retrieved by the retriever often contains a significant amount of noise, this interferes with the LLM's ability to provide accurate answers.
>
> [1] Zhang, X., Song, Y., Wang, Y., Tang, S., Li, X., Zeng, Z., ... & Wen, Q. (2024). RAGLAB: A Modular and Research-Oriented Unified Framework for Retrieval-Augmented Generation.
>
> [2] Hexiang Tan, Fei Sun, Wanli Yang, Yuanzhuo Wang, Qi Cao, and Xueqi Cheng. 2024. Blinded by Generated Contexts: How Language Models Merge Generated and Retrieved Contexts When Knowledge Conflicts?
>
> We hope our clarifications address your concerns and answer your questions. Thank you once again for your valuable insights.

---

> ### Comment · Reviewer_kEed · 2024-11-24
>
> Thank you for taking the time to address my concerns.
> - I am satisfied with the SFT analysis, thank you for providing it.
> - I appreciate the thorough description of the "preference gap" and how your framework addresses it. I believe training a model to optimize what a retriever returns and what is ultimately presented to an LLM based on the LLM's own preferences is an important direction to explore.
>
> However, I believe it is still not ready for acceptance based on 2 issues:
> 1) The example provided in response to the 6th concern of reviewer m95q makes it seem like the main improvements for GasketRAG come from its iterative retrieval ability, not the "preference gap" that motivates its design. I think more explicit examples of such a gap would be necessary to provide a more convincing story.
> 2) Similarly, I mentioned before that NaiveRAG's underperformance against direct prompting on 2Wiki was unexpected. However, upon further inspection, your experiments show that direct prompting outperforms all iterative retrieval methods such as Iter-RetGen and Self-Ask even though they have been shown to work in their own papers and others [1][2][3]. Why is it that simpler iterative retrieval methods provide no improvements in your setting? Considering its limited complexity, I think understanding this is a necessary step in truly understanding the benefits of GasketRAG.
>
> I have changed my overall score from 3 to 5 and the Contribution score from 2 to 3.
>
> [1] https://arxiv.org/pdf/2212.10509
>
> [2] https://arxiv.org/pdf/2405.14831
>
> [3] https://arxiv.org/pdf/2410.04343

---

> > ### Author Response · Authors · 2024-11-25
> >
> > Thank you for your response and for continuing our discussion. We greatly appreciate your insightful and professional feedback. Below are our responses to the new issues you have raised.
> >
> > **Response to Issue 1:**
> >
> > The main purpose of the example we presented to Reviewer M95Q is to demonstrate how the two iterations of GasketRAG help the LLM correctly answer the question. This example is highly intuitive and relatively easy to understand from a human perspective. We agree with your observation that the improvement in this example is attributed to the iterative retrieval ability. However, this precisely reflects **how the retriever preference is learned by the gasket model:** the original query fails to retrieve useful information, so **the gasket model, leveraging the retriever's processing habits it has learned, enhances the query in a targeted manner**, enabling it to rank useful information higher. It is worth noting that this example only reflects one of the most easily understood aspects of the gasket model. In fact, **the preferences of the retriever and LLM resemble a black box, making the underlying mechanisms sometimes difficult to interpret in a way that is easily comprehensible to humans.** To further address your question, we present the following example:
> >
> > ```json
> > {
> >         "Query": "When was the director of film Jinpa born?",
> >         "Iter-1 sentences ": [
> >             "[s_14] \"Jinpa Jinpa () is a 2018 Chinese Tibetan-language film written and directed by Pema Tseden.",
> >             "[s_17] Truck driver Jinpa (Jinpa) accidentally runs over and kills a sheep as he traverses on an isolated road on the Kekexili Plateau.",
> >             "[s_23] Jinpa Jinpa () is a 2018 Chinese Tibetan-language film written and directed by Pema Tseden."
> >         ],
> >
> >         "Iter-1 answer": "The director of film Jinpa was born in 1970.",
> >         "Iter-2 sentences": [
> >             "[s_1] \"Jinpa Jinpa () is a 2018 Chinese Tibetan-language film written and directed by Pema Tseden.",
> >             "[s_10] Jinpa Jinpa () is a 2018 Chinese Tibetan-language film written and directed by Pema Tseden.",
> >             "[s_12] It made its world premiere and won Best Screenplay in the Horizons section at the 75th edition of the Venice Film Festival."
> >         ],
> >         "Iter-2 answer": "Pema Tseden was born in 1969.",
> >         "NaiveRAG answer": "May 25, 1955",
> >         "Iter-RetGen answer": "1955",
> >         "Direct answer": "The director of film Jinpa, Pema Tseden, was born in 1970.",
> >         "true_answer": "1969"
> >  }
> > ```
> >
> > For this question, NaiveRAG, Iter-RetGen, Direct Prompt, and the first round of GasketRAG all generated hallucinations in their responses. Although the retriever's results included background information about the movie *Jinpa* and its director, Pema Tseden, they did not provide Pema Tseden's birth year. It can be observed that, even though the sentences provided to the LLM in both rounds of the Gasket model did not contain the correct answer, GasketRAG was still able to correctly answer the question in the second round. To further study, We directly asked the LLM the following question:
> >
> > ```json
> > Prompt: "When was Pema Tseden born?"
> > LLM Response: "1969"
> > ```
> >
> > This indicates that the LLM's parameter knowledge contains the correct answer to the question. However, due to the **LLM's extreme sensitivity to specific noise in the context**, it fails to generate an answer it should inherently know (even Direct Prompt contains noise). **The Gasket model, through its two rounds of adjustment enables the LLM to unlock its parameter knowledge when the retriever fails to provide the answer.** This process may not be interpretable in a way that aligns with human understanding, highlighting the necessity of using GasketRAG to bridge the preference gap. This is because defining simple rules to align the retriever and the LLM is impossible.

---

> > > ### Author Response · Authors · 2024-11-25
> > >
> > > **Response to Issue 2:**
> > >
> > > The cases presented above also help explain why Direct Prompt sometimes performs even better than other RAG methods. **When the retriever's results do not contain the correct answer, the LLM tends to rely on the provided context rather than utilizing its parameter knowledge, which often leads to hallucinations.** In GasketRAG, the gasket model facilitates alignment between the LLM and the retriever, enabling the LLM to adapt to the varying quality of the retriever's outputs.
> > >
> > > We present one more example:
> > >
> > > ```json
> > > {
> > >         "Query": "Did the movies Passer L'Hiver and In My Skin, originate from the same country?",
> > >         "Iter-1 sentences ": [
> > >             "[s_1] \"Skin (2018 film) Skin is an American biographical drama film written and directed by Guy Nattiv, based on his short film of the same name.",
> > >             "[s_2] It follows the life of former skinhead group member Bryon Widner, and stars Jamie Bell, Vera Farmiga, Danielle Macdonald, Mike Colter, and Bill Camp.",
> > >             "[s_29] In The Skin I Live In () is a 2011 Spanish psychological horror film written and directed by Pedro Almod\u00f3var, starring Antonio Banderas, Elena Anaya, Marisa Paredes, Jan Cornet, and Roberto \u00c1lamo.",
> > >             "[s_46] In The Skin I Live In () is a 2011 Spanish psychological horror film written and directed by Pedro Almod\u00f3var, starring Antonio Banderas, Elena Anaya, Marisa Paredes, Jan Cornet, and Roberto \u00c1lamo."
> > >         ],
> > >
> > >         "Iter-1 answer": "No, the movies Passer L'Hiver and In My Skin did not originate from the same country.",
> > >         "Iter-2 sentences": [],
> > >         "Iter-2 answer": "Yes, both movies originated from France.",
> > >         "NaiveRAG answer": "The movies Passer L'Hiver and In My Skin did not originate from the same country.",
> > >         "Iter-RetGen answer": "No, the movies Passer L'Hiver and In My Skin did not originate from the same country.",
> > >         "Direct answer": "Yes, both movies originated from France.",
> > >         "true_answer": "yes"
> > >  }
> > > ```
> > >
> > > In this example, it can be observed that in the second round, the gasket model chose not to select any sentences at all because all retrieved results were useless to the question. As a result, the LLM, unaffected by the noisy context, relied on its internal knowledge to correctly answer the question. In contrast, other RAG methods misled the LLM.
> > >
> > > The results differences compared to the papers you pointed out may stem from the use of different corpora. In our experiments, we utilized the WiKi2018 corpus published by the DPR [1] project. **Considering that retrievers often fail to provide ideal retrieval results in real-world scenarios, our research is of significant importance for improving the stability of RAG systems** under such conditions.
> > >
> > > Thank you once again for your thorough review. We hope our responses address your concerns. We will also actively respond to any new questions you may have.
> > >
> > > [1] Karpukhin, V., Oğuz, B., Min, S., Lewis, P., Wu, L., Edunov, S., ... & Yih, W. T. (2020). Dense passage retrieval for open-domain question answering.

---

> > > > ### Comment · Reviewer_kEed · 2024-11-25
> > > >
> > > > Thank you for the examples, they illustrate some of the advantages of GasketRAG in a bit more detail. However, I will leave my score as is.
> > > >
> > > > Since this paper's main contribution is the improvements obtained by GasketRAG, I believe that more analysis should be provided to characterize its pros and cons. From the 3 examples the authors provided in the rebuttal, 3 improvement categories can already be seen such as 1) iterative retrieval, 2) re-ranking and 3) retrieval ablation. I would like to see a deeper dive into these categories as well as some error analysis since GasketRAG must have some limitations and they should be discussed.

---

> > > > > ### Author Response · Authors · 2024-11-25
> > > > >
> > > > > Thank you for your feedback. Through our experiments and case studies, we demonstrated the effectiveness of GasketRAG from both quantitative and qualitative perspectives, and these results are sufficient to support our statements. Forcing all the improvements of GasketRAG into the categories you mentioned would be unreasonable because the preference gap between the retriever and the LLM arises precisely from their inconsistency with human preferences. Thus, **attempting to explain all the behaviors of the gasket model using human understanding contradicts the motivation behind GasketRAG**. LLMs are very possible to infer incorrect answers due to seemingly “helpful” context or, conversely, arrive at correct answers triggered by apparently “unrelated” background knowledge.
> > > > >
> > > > > We recommend understanding GasketRAG in this way: the gasket model learns how the LLM and the retriever react to various inputs with noise, and thereby fits the optimal choices to guide the entire system toward the correct objective. While many studies aim to explain LLMs, the explanations of how LLMs react to noise are still very limited. Furthermore, research on LLM interpretability is clearly beyond the scope of this paper.
> > > > >
> > > > > Regarding the limitations, we believe that using larger, more powerful LLMs as the base model could bring potential improvements. In Section 4.6, with a smaller 1.5B model and less training data, our method still achieved significant improvements. Moreover, our method does not require complex joint training and features a streamlined framework. **GasketRAG is highly efficient and easy to operate from both training and inference perspectives.** Considering these factors, GasketRAG demonstrates high practical value, and we have also made our code publicly available. Therefore, we hope that this work will gain the recognition.
> > > > >
> > > > > Thank you once again for your response. Wishing you a wonderful day!

---

> > > > > > ### Comment · Reviewer_kEed · 2024-11-29
> > > > > >
> > > > > > I have been thinking about it and I don't agree with the author's comment "attempting to explain all the behaviors of the gasket model using human understanding contradicts the motivation behind GasketRAG". I believe that any scientific paper which provides empirical improvements needs to properly explain and characterize its improvements. Simply saying that it addresses the "preference gap" is not enough, especially since the examples that the authors gave were mostly due to other reasons like iterative retrieval improvements. Defining qualitative analysis as "LLM interpretability" and assuming it is out of the scope is not acceptable from my perspective.
> > > > > >
> > > > > > I will change my score back to 3.

---

> > > > > > > ### Author Response · Authors · 2024-11-30
> > > > > > >
> > > > > > > We appreciate the your meticulous approach to the review process. To avoid our discussion devolving into a tug-of-war over subjective opinions, we have analyzed the improvements of GasketRAG as per your request. First, we will present these analyses, and then address the following points:
> > > > > > >
> > > > > > > 1. **Why we believe it is unreasonable to demand a human-level understanding of the preferences of LLMs and retrievers.**
> > > > > > > 2. **The misunderstandings in your interpretation of the sources of improvement.**
> > > > > > >
> > > > > > > We manually analyzed the 17 instances where GasketRAG's responses outperformed  both of NaiveRAG and Iter-RetGen on the WikiMultiHop test set (first 500). In addition to the three categories you listed—Iterative Retrieval, Reranking, and Retrieval Ablation—we have introduced three additional categories: Indirect Hint, Partial Background Hidden, and Noise Manipulation.
> > > > > > >
> > > > > > > Explanations are as follows:
> > > > > > >
> > > > > > > - **Indirect Hint**: The sentences output by the Gasket model to the LLM contain some background information but do not include the answer to the question.
> > > > > > > - **Partial Background Hidden**: The Gasket model deliberately conceals part of the background information to prompt the LLM to produce the desired answer.
> > > > > > > - **Noise Manipulation**: The model manipulates completely unrelated background information but successfully guides the LLM to answer the question correctly.
> > > > > > >
> > > > > > > Among the 17 samples, the distribution of cases across the various categories is as follows:
> > > > > > >
> > > > > > > ```python
> > > > > > > Indirect Hint 4
> > > > > > > Partial Background Hidden 4
> > > > > > > Iterative Retrieval 4
> > > > > > > Reranking 1
> > > > > > > Retrieval Ablation 2
> > > > > > > Noise Manipulation 2
> > > > > > > ```
> > > > > > >
> > > > > > >  The following is an example of a **Partial Background Hidden** case.
> > > > > > >
> > > > > > > **Case 1:**
> > > > > > >
> > > > > > > ```python
> > > > > > >     {
> > > > > > >         "query": "Which film was released earlier, Moment Of Danger or The Ballad Of Josie?",
> > > > > > >         "Iter-1 sentences": [
> > > > > > >             "[s_17] \"Moment of Danger Moment of Danger (also known as Malaga) is a 1960 crime drama film starring Trevor Howard, Dorothy Dandridge and Edmund Purdom.",
> > > > > > >             "[s_1] \"The Ballad of Josie The Ballad of Josie is a 1967 Technicolor American comedy western film directed by Andrew V. McLaglen and starring Doris Day, Peter Graves and George Kennedy.",
> > > > > > >             "[s_10] The Ballad of Josie The Ballad of Josie is a 1967 Technicolor American comedy western film directed by Andrew V. McLaglen and starring Doris Day, Peter Graves and George Kennedy."
> > > > > > >         ],
> > > > > > >         "Iter-1 answer": "The Ballad of Josie",
> > > > > > >         "Iter-2 sentences": [
> > > > > > >             "[s_1] \"Moment of Danger Moment of Danger (also known as Malaga) is a 1960 crime drama film starring Trevor Howard, Dorothy Dandridge and Edmund Purdom.",
> > > > > > >             "[s_4] The film proved to be the final completed film for Dorothy Dandridge.",
> > > > > > >             "[s_5] Starting with a wordless jewel heist pulled-off by thief Peter Curran and locksmith John Bain, Curran then double-crosses his accomplice, dumps his lover Gianna and escapes with his ill-gotten gains.",
> > > > > > >             "[s_10] It was filmed in Europe in the late months of 1959.",
> > > > > > >             "[s_12] The film proved to be the final completed film for Dorothy Dandridge."
> > > > > > >         ],
> > > > > > >         "Iter-2 answer": "Moment of Danger",
> > > > > > >         "NaiveRAG answer": "The Ballad of Josie was released earlier than Moment of Danger.",
> > > > > > >         "Iter-RetGen answer": "The Ballad of Josie",
> > > > > > >         "Direct answer": "The Ballad Of Josie was released earlier.",
> > > > > > >         "true_answer": "Moment Of Danger",
> > > > > > >         "explain": "Partial Background Hidden"
> > > > > > >     },
> > > > > > > ```
> > > > > > >
> > > > > > > In this case, it can be observed that although in the first round GasketRAG's gasket model accurately provided the LLM with evidence that could have easily led to the correct answer, the LLM still answered incorrectly. Similarly, NaiveRAG, Iter-RetGen, and Direct all failed to provide the correct answer. However, in the second round, a reversal occurred: the gasket model removed all information related to *The Ballad of Josie* and successfully guided the LLM to answer the question correctly.

---

> > > > > > > ### Author Response · Authors · 2024-11-30
> > > > > > >
> > > > > > > The following is an example of an **Indirect Hint** case.
> > > > > > >
> > > > > > > **Case 2:**
> > > > > > >
> > > > > > > ```python
> > > > > > >     {
> > > > > > >         "query": "Where was the director of film Ronnie Rocket born?",
> > > > > > >         "Iter-1 sentences": [
> > > > > > >             "[s_0] Title: \"Ronnie Rocket\".",
> > > > > > >             "[s_12] \"Ronnie Rocket Ronnie Rocket is an unfinished film project written by David Lynch, who also intended to direct it.",
> > > > > > >             "[s_13] Begun after the success of Lynch's 1977 film \"\"Eraserhead\"\", \"\"Ronnie Rocket\"\" was shelved after Lynch felt he would be unable to find financial backing for the project.",
> > > > > > >             "[s_34] After releasing 1977's \"\"Eraserhead\"\", a black-and-white surrealist film and his début feature-length production, Lynch began work on the screenplay for \"\"Ronnie Rocket\"\".",
> > > > > > >             "[s_30] The project has also suffered setbacks due to the bankruptcy of several potential backers; both Dino De Laurentiis's De Laurentiis Entertainment Group and Francis Ford Coppola's American Zoetrope were attached to the project at different times; both production companies went bankrupt before work could begin."
> > > > > > >         ],
> > > > > > >         "Iter-1 answer": "David Lynch",
> > > > > > >         "Iter-2 sentences": [
> > > > > > >             "[s_1] \"Ronnie Rocket Ronnie Rocket is an unfinished film project written by David Lynch, who also intended to direct it.",
> > > > > > >             "[s_49] Title: \"Early life of David Lynch\"."
> > > > > > >         ],
> > > > > > >         "Iter-2 answer": "David Lynch was born in Missoula, Montana.",
> > > > > > >         "NaiveRAG answer": "Ronnie Rocket was born in Bismarck, North Dakota, United States.",
> > > > > > >         "Iter-RetGen answer": "Bismarck, North Dakota",
> > > > > > >         "Direct answer": "The director of film Ronnie Rocket was born in New York City, New York, United States.",
> > > > > > >         "true_answer": "Missoula, Montana",
> > > > > > >         "explain": "Indirect Hint"
> > > > > > >     },
> > > > > > > ```
> > > > > > >
> > > > > > > In this case, the gasket model provided the name of Ronnie Rocket's director, David Lynch, in the first round of selected sentences. However, the LLM was clearly unable to provide a correct answer. In the second round, the gasket model still did not provide David Lynch's specific birthplace but adjusted some cues: "[s_49] Title: 'Early life of David Lynch'." This activated the LLM's parameter knowledge, leading to the correct answer. **The specific reason why the gasket's output in the first round led it to make such a choice in the second round is likely difficult to interpret from a human perspective. This is precisely because the *preference gap* between the retriever and the LLM is not readily understandable to humans**. It is **this lack of interpretability** that **motivates the use of the gasket RAG framework** to bridge this gap and align the retriever's outputs with the LLM's preferences.

---

> > > > > > > ### Author Response · Authors · 2024-11-30
> > > > > > >
> > > > > > > Finally, here is an example of a **Noise Manipulation** case. This might seem more like magic.
> > > > > > >
> > > > > > > **Case 3:**
> > > > > > >
> > > > > > > ```python
> > > > > > >     {
> > > > > > >         "query": "Which film has the director who was born earlier, Retalhos Da Vida De Um Médico or The Shooting?",
> > > > > > >         "Iter-1 sentences": [
> > > > > > >             "[s_1] \"\"\"Fogo na Noite Escura\"\" (1943), at the collection \"\"Novos Prosadores\"\" (1943), by \"\"Coimbra Editora\"\".",
> > > > > > >             "[s_2] Besides over 30 titles, along his fifty years of intensive literary life, not only wrote “neo-realistic” novels, as \"\"Casa da Malta\"\" (1945), \"\"Minas de S. Francisco\"\" (1946), \"\"Retalhos da Vida de um Médico\"\" (1949 and 1963), \"\"A Noite e a Madrugada\"\" (1950), \"\"O Trigo e o Joio\"\" (1954), but also “urban themes”, contemporary fiction, as in \"\"O Homem Disfarçado\"\" (1957), \"\"Cidade Solitária\"\" (1959), \"\"Domingo à Tarde\"\" (1961, José Lins do Rego Prize), \"\"Os Clandestinos\"\" (1972), \"\"Resposta a Matilde\"\" (1980) or \"\"O Rio Triste\"\" (1982, Fernando Chinaglia\"",
> > > > > > >             "[s_3] Title: \"Autran Dourado\".",
> > > > > > >             "[s_4] \"in the Portuguese \"\"sprachraum\"\".",
> > > > > > >             "[s_5] In 2001, Brazilian filmmaker Suzana Amaral released the film \"\"Uma Vida em Segredo\"\".",
> > > > > > >             "[s_6] It was based on the novel of same title by Autran Dourado."
> > > > > > >         ],
> > > > > > >         "Iter-1 answer": "The Shooting",
> > > > > > >         "Iter-2 sentences": [
> > > > > > >             "[s_2] Besides over 30 titles, along his fifty years of intensive literary life, not only wrote “neo-realistic” novels, as \"\"Casa da Malta\"\" (1945), \"\"Minas de S. Francisco\"\" (1946), \"\"Retalhos da Vida de um Médico\"\" (1949 and 1963), \"\"A Noite e a Madrugada\"\" (1950), \"\"O Trigo e o Joio\"\" (1954), but also “urban themes”, contemporary fiction, as in \"\"O Homem Disfarçado\"\" (1957), \"\"Cidade Solitária\"\" (1959), \"\"Domingo à Tarde\"\" (1961, José Lins do Rego Prize), \"\"Os Clandestinos\"\" (1972), \"\"Resposta a Matilde\"\" (1980) or \"\"O Rio Triste\"\" (1982, Fernando Chinaglia\"",
> > > > > > >             "[s_1] \"\"\"Fogo na Noite Escura\"\" (1943), at the collection \"\"Novos Prosadores\"\" (1943), by \"\"Coimbra Editora\"\"."
> > > > > > >         ],
> > > > > > >         "Iter-2 answer": "Retalhos Da Vida De Um Médico",
> > > > > > >         "NaiveRAG answer": "The director of The Shooting was born earlier than the director of Retalhos Da Vida De Um Médico.",
> > > > > > >         "Iter-RetGen answer": "The Shooting",
> > > > > > >         "Direct answer": "The Shooting",
> > > > > > >         "true_answer": "Retalhos Da Vida De Um Médico",
> > > > > > >         "Explain": "Noise Manipulation"
> > > > > > >     },
> > > > > > > ```
> > > > > > >
> > > > > > > In this case, the gasket model did not provide any sentences related to the two films. In the first round, GasketRAG, like other methods, also answered incorrectly. However, in the second round, the gasket model's manipulation of the noise altered the LLM's response, leading to the correct answer.

---

> > > > > > > ### Author Response · Authors · 2024-11-30
> > > > > > >
> > > > > > > Having presented these cases, we will now articulate our viewpoints.
> > > > > > >
> > > > > > > **Point 1: Attempting to explain all the behaviors of the gasket model using human understanding contradicts the motivation behind GasketRAG.**
> > > > > > >
> > > > > > > From the **above cases**, it is evident that **many of the gasket model's operations are conducted in ways that are not readily comprehensible to humans.** These operations stem from the gasket model's understanding of the characteristics of the LLM and retriever, which it gained through preference learning. This cannot simply be attributed to iterative retrieval or reranking, as the results of GasketRAG are clearly superior to those of Iter-RetGen. Even in cases where the retriever successfully retrieved the correct answer, the LLM sometimes failed to produce the correct response due to its inherent limitations. The gasket model, by understanding some of these underlying characteristics of the LLM, was able to guide it toward generating the correct answer.  **Together with our paper and rebuttal, we have made our best effort to analyze the improvements of GasketRAG.**
> > > > > > >
> > > > > > > **Point 2: The reviewer’s attribution of the improvement of GasketRAG has issue.**
> > > > > > >
> > > > > > > Attributing the improvements brought by GasketRAG to Iterative Retrieval，Re-ranking, while attempting to isolate the elimination of the preference gap from these two factors, is problematic. **Iterative Retrieval and Re-ranking are operations, tools employed to achieve the goal of eliminating the preference gap.** If following the reviewer’s thought, Retrieval Ablation could also be classified as Re-ranking, and nearly all improvements would then be attributed to Iterative Retrieval and Re-ranking. Extending this further, you might even critique Partial Background Hidden, Noise Manipulation, Indirect Hint, or any improvement source we propose as merely forms of Iterative Retrieval and Re-ranking. This would lead our discussion into a deadlock.
> > > > > > >
> > > > > > > Based on our experiments, the advantages of GasketRAG over Iter-RetGen and other RAG methods are objective and significant. **GasketRAG's Iterative Retrieval and Re-ranking differ fundamentally from those in other methods, with its core innovation lying in how it addresses the preference gap.**
> > > > > > >
> > > > > > > Finally, **we want to clarify that the attribution analysis provided above follows the direction suggested by the reviewer, with the aim of offering more details to help anyone with similar questions understand the mechanisms behind GasketRAG.** We will also include the case study in the paper. However, **this does not imply that we agree with the necessity of the attribution analysis or that we consider the lack of such an analysis a major flaw in our paper.** **We also do not claim that the improvements brought by GasketRAG are limited to the above attributions.** Since attribution classification involves subjective interpretation, we also anticipate that reviewer may have differing perspectives.
> > > > > > >
> > > > > > > We sincerely thank you for your diligence and dedication. Our community greatly benefits from reviewers as engaged and thoughtful as you. While we recognize that there may be differences in perspective, we genuinely appreciate the insights you have provided. If we have addressed your concerns, we kindly ask you to reconsider your evaluation of our work.

---

> > > > > > > > ### Comment · Reviewer_kEed · 2024-11-30
> > > > > > > >
> > > > > > > > Thank you for performing and adding the analysis to the paper!
> > > > > > > >
> > > > > > > > I don't quite understand the author's perspective on human interpretations of GasketRAG's improvements or limitations being unimportant. I understand that GasketRAG's motivation was not to directly address these problems but characterizing its behavior leads to not only a deeper understanding of how your system can be used in practice but **also** what can be learned through minimizing the preference gap directly (which I believe is actually even more interesting). Including qualitative examples from different datasets and characterizing them also makes the whole paper more convincing, especially given the unexpected finding that RAG methods underperform Direct generation.
> > > > > > > >
> > > > > > > > Overall, however, I believe the authors have made a thorough effort to address my concerns despite our disagreements. Although I would love to see more analysis on the improvements on the other datasets I believe as long as the analysis done above is included in the paper I will raise my score from 3 to 6.

---

> > > > > > > > > ### Author Response · Authors · 2024-12-01
> > > > > > > > >
> > > > > > > > > Thank you for recognizing our efforts. We do not consider human interpretations of GasketRAG's improvements or limitations to be unimportant. Our concern lies in the fact that when it becomes challenging to track and interpret the gasket model's processing of each token, the attribution for these individual cases often involves a high degree of subjectivity. As a result, such interpretations could lead to unnecessary controversy. However, we agree with your point that analyzing the behavior of the gasket model is indeed valuable for understanding what can be learned through directly minimizing the preference gap.
> > > > > > > > >
> > > > > > > > > We additionally analyzed the cases on HotPotQA and PopQA. Since we are currently unable to update the PDF on OpenReview, we have uploaded the revised paper to our anonymous repository: https://anonymous.4open.science/r/9668 You can find the PDF file in the root directory. We have updated the case study content in Appendix A.
> > > > > > > > >
> > > > > > > > > Thank you again for your active discussion.

---

### Official Review · Reviewer_m95q · 2024-11-03

**Soundness:** 2
**Presentation:** 2
**Contribution:** 2
**Rating:** 6
**Confidence:** 3

**Summary:**

The paper introduces GasketRAG, a novel approach designed to enhance Retrieval-Augmented Generation (RAG) by optimizing the interaction between retrievers and large language models (LLMs). GasketRAG incorporates a "gasket" mechanism that facilitates improved collaboration between the retriever and LLM. This approach allows for the simultaneous optimization of both retrieval ranking and document refinement, eliminating the complexities associated with traditional training and inference pipelines. In comparative tests against state-of-the-art RAG methods, GasketRAG demonstrated superior performance across multiple datasets.

**Strengths:**

1.	This paper proposes a method to use an enhanced query (query + important sentences) to improve the performance of RAG techniques.

2.	This method considers both the preference of LLMs and retrievers by training gasket with data integrated with these preferences.

3.	Experiment performances show the effectiveness of the method.

**Weaknesses:**

1.	The motivation of the Gasket remains unclear to me. The authors claims that gasket is designed to consider the preference of both the retrievers and the LLM and that it serves as an information bottleneck to control the behavior of both the retriever and the LLM. However, it’s difficult to understand why retrieving the documents again with a combination of query and retrieved sentences could necessarily “control” the behavior of the retriever, consider its preference and benefit the final prediction. As shown in Table 4, more iterations could even harm the performance.

2.	Also, the claim about the alignment between LLMs and retrievers is also questionable. The only related part in the paper is section 3.3, where the training data of the gasket is collected considering the preference of LLMs and retrievers. However, it remains unclear how and why this could align them.

3.	The method could not outperform the baseline models if it’s not specifically trained on the target dataset (WikiMultiHop, PubHealth, and StrategyQA). Also, I find the setting of testing on the dataset that has been used for the training of gasket is not quite fair for other baseline models since they do not require this.

4.	The overall logic of the paper could be improved.

**Questions:**

1.	I think you should use \text{-} in the equations from L161-L168, now it looks like Top “minus” k.

2.	Could you explain a little bit more about the definition of “retriever preference” and how it’s related to the overall inference period of the proposed method?

3.	Are there analyses about the differences between the sentences selected in the first and the second round?

4.	Are there analyses about the advantage of the sentences retrieved in the second round and how could we explain the results in a more concrete way than only saying it’s an alignment between the preference of LLMs and retrievers?

---

> ### Author Response · Authors · 2024-11-16
>
> Thank you for your thoughtful review and feedback on our submission. We appreciate the time and effort you invested in understanding our work. We would like to address the concerns you raised:
>
> **Concern 1: Motivation of GasketRAG and how to understand the preferences of the retriever and the LLM.**
>
> **Response:**
>
> Retrievers are generally trained based on human preferences, designed to retrieve and rank documents in a way that aligns with human habits. However, the preferences of LLMs do not completely align with those of humans. As pointed out in [1], when LLMs are presented with both LLM-generated and retrieved documents, they tend to favor content generated by LLMs. This demonstrates subtle differences in how LLMs process contexts from different sources. Furthermore, [2] validates that LLMs are not particularly sensitive to the ranking of documents but are highly sensitive to irrelevant information (noise) within documents. **We define the divergence between the retriever's output preferences and the LLM's input preferences as the *preference gap*.** This means that the direct output of the retriever is not ideal as input for the LLM. Therefore, we establish a gasket model through preference training to filter information from the retrieved documents, aligning more closely with the LLM's input preferences to help it generate accurate responses. At the same time, the output of the gasket model is also used to enhance the query fed to the retriever, thereby adjusting the retriever's behavior and improving its retrieval results.
>
> To give **a straightforward example**, for certain queries, an LLM might possess parameter knowledge and can provide answers without requiring additional context. In such cases, the content retrieved by the retriever might have a negative impact. The gasket model learns these preferences from training data and tends to avoid selecting any sentence from the retrieved passages to reduce interference when it believes the LLM can answer the question without assistance. Compared to the preferences of the LLM, the **preferences of the retriever** are actually easier to understand. The original question is not always correctly interpreted by the retriever, so the retrieval results are often suboptimal. Therefore, the gasket model also needs to enhance the input to the retriever to help it generate LLM preferred retrieval results.
>
> The explanation above is provided for a more intuitive understanding; in reality, the input-output dynamics of retrievers and LLMs are far more complex. This complexity makes the preference gap between retrievers and LLMs resemble a black box. By designing methods to collect preference data, we enable the gasket model to **learn the collaborative patterns** between the retriever and the LLM, thereby understanding their preferences. Acting as a coordinator, the gasket model maximizes the performance of the entire RAG system.
>
> [1] Hexiang Tan, Fei Sun, Wanli Yang, Yuanzhuo Wang, Qi Cao, and Xueqi Cheng. 2024. Blinded by Generated Contexts: How Language Models Merge Generated and Retrieved Contexts When Knowledge Conflicts?
>
> [2] Zixuan Ke, Weize Kong, Cheng Li, Mingyang Zhang, Qiaozhu Mei, and Michael Bendersky. 2024. Bridging the Preference Gap between Retrievers and LLMs.
>
> ---
>
> **Concern 2: Why retrieving the documents again with a combination of query and retrieved sentences could necessarily “control” the behavior of the retriever?**
>
> **Response:**
>
> During training, the gasket model learns how to enhance queries by observing the impact of modified input of the retiever on the final result (LLM's answer). This process helps the gasket model guide the retriever to retrieve the desired documents. Therefore, this is a result-oriented approach. We do not focus on how the gasket model improves the retriever; we only reward the gasket model when it performs the correct actions (selected the right sentences). Thus, the preferences of the retriever are implicitly learned.
>
> ---
>
> **Concern 3: As shown in Table 4, more iterations could even harm the performance.**
>
> **Response:**
>
> Although the gasket model partially bridges the gap between the retriever and the LLM, it is not perfect. During its sentence selection process, it may still introduce irrelevant information, which we refer to as noise. The introduction of noise can potentially shift the retriever's retrieval direction, and this shift may be amplified with more iterations. Therefore, more iterations do not necessarily lead to better results. Similar phenomena can also be observed in other iterative RAG methods.

---

> ### Author Response · Authors · 2024-11-16
>
> **Concern 4: Why the preference data could align the retriever and the LLM?**
>
> **Response:**
>
> We understand that our explanation of preference training might seem complex, so here’s a simpler way to explain it:
>
> First, let's consider only the first iteration of GasketRAG. This represents a straightforward RAG structure consisting of a retriever, a gasket (acting as a refiner), and an LLM. By applying preference training to the gasket, we align the entire pipeline with the LLM's accuracy in generating the final answer. However, this approach has limitations because the retriever's input and output remain unchanged.
>
> To address this, we aim to adjust the retriever's input and output as well. Instead of introducing an additional model, we continue using the gasket model to enhance the retriever's input query. Consequently, the gasket model's output is constrained by **two objectives**:
>
> 1. **Ensuring the LLM produces accurate answers.**
> 2. **Guiding the retriever to retrieve desired documents.**
>
> The question now becomes how to **define "more desired" retrieval results** for the retriever. We approach this in two scenarios:
>
> 1. **First-Round Incorrect Answer, Second-Round Correct Answer**
>
>     If the sentence list selected by the gasket model in the first round led the LLM to generate an incorrect answer, but after a second round of retrieval and sentence selection, the LLM produces the correct answer, we consider the retrieval in the second round as "more desired." This means the first-round output of GasketRAG effectively guided the retriever to improve its results.
>
> 2. **First-Round Correct Answer**
>
>     If the LLM already generated the correct answer in the first round, it will also produce a correct answer in the second round. In this case, we evaluate the average index of the sentences selected by the gasket model in the two rounds. If the average index decreases compared to the first round, it indicates that relevant information has been moved closer to the top of the retrieval results. Such a retrieval is also considered "more desired."
>
>
> Using these criteria, we label the sentence lists produced by the gasket model as **strong preferred, weak preferred, strong dispreferred or weak dispreferred** for training. This labeling allows the gasket model to learn and optimize towards achieving the **two objectives**.
>
> ---
>
> **Concern 5: The comparison between GasketRAG and baseline methods.**
>
> **Response:**
>
> Our experimental results demonstrate the significant effectiveness of the GasketRAG method. As shown in Table 2, GasketRAG (LLaMA-3) outperforms other methods by a large margin on the PopQA, TriviaQA, HotpotQA, and WikiMultiHop test sets, and is very close to the best-performing method on the PubHealth and StrategyQA datasets. Moreover, it exhibits consistent performance on untrained OOD (Out-of-Domain) test sets. We observed that, unlike GasketRAG, other RAG methods perform significantly worse than NaiveRAG on some datasets, highlighting the superior **stability** of GasketRAG.
>
> It is worth noting that our gasket model was trained using preference-aligned data specific to LLaMA-3. However, even when the generator LLM was replaced with GPT-3.5, GasketRAG still achieved superior performance on multiple datasets, demonstrating the excellent **generalization capability** of our approach.
>
> Our primary evaluation metric is **Correctness**. Given the free-form nature of LLM outputs, correctness as judged by GPT-4 based on semantic understanding is more convincing compared to accuracy (ACC).

---

> ### Author Response · Authors · 2024-11-16
>
> **Concern 6: Analyses of the sentences selected.**
>
> **Response:**
>
> Below, we present an example of GasketRAG inference. In the first round of retrieval, the retriever, based on the original query, identified that Millett is the author of *Sexual Politics*. However, the returned documents lacked information that could truly help answer the question. As a result, the sentences selected by the gasket model could not enable the LLM to produce the correct answer.
>
> However, these selected sentences helped the retriever locate a critical fragment, "Millett attended Oxford University," during the second round of retrieval. Additionally, we observe that in the first round, the selected sentences had indices s_8, s_10, and s_27, while in the second round, they were s_1, s_2, s_3, and s_10. The sentences selected in the second round clearly had lower indices, indicating that the edits made by the gasket model to the query effectively surfaced relevant information in the retrieval results.
>
> ```json
> {
>         "Query": "The author of Sexual Politics attended which British University?",
>         "Iter-1 sentences ": [
>             "[s_8] \"Kate Millett Katherine Murray Millett (September 14, 1934 \u2013 September 6, 2017) was an American feminist writer, educator, artist, and activist.",
>             "[s_10] She has been described as \"\"a seminal influence on second-wave feminism\"\", and is best known for her book \"\"Sexual Politics\"\" (1970), which was based on her doctoral dissertation at Columbia University.",
>             "[s_27] \"\"Sexual Politics\"\" originated as Millett's PhD dissertation and was published in 1970, the same year that she was awarded her doctorate from Columbia University."
>         ],
>
>         "Iter-1 answer": "Columbia University",
>         "Iter-2 sentences": [
>             "[s_1] \"Kate   Katherine Murray Millett (September 14, 1934 \u2013 September 6, 2017) was an American feminist writer, educator, artist, and activist.",
>             "[s_2] She attended Oxford University and was the first American woman to be awarded a degree with first-class honors after studying at St Hilda's College, Oxford.",
>             "[s_3] She has been described as \"\"a seminal influence on second-wave feminism\"\", and is best known for her book \"\"Sexual Politics\"\" (1970), which was based on her doctoral dissertation at Columbia University.",
>             "[s_10] She attended Oxford University and was the first American woman to be awarded a degree with first-class honors after studying at St Hilda's College, Oxford."
>         ],
>         "Iter-2 answer": "Oxford University",
>         "true_answer": "Oxford"
>  }
> ```
>
> ---
>
> **Concern 7: Typos and presentations.**
>
> **Response:**
>
> Thank you very much for your thorough review of our paper. We will promptly correct the typos you pointed out and refine our phrasing.
>
> We hope our clarifications address your concerns and answer your questions. Thank you once again for your valuable insights.

---

> ### Author Response · Authors · 2024-11-26
>
> Dear reviewer,
>
> I hope this message finds you well.
>
> We wanted to kindly follow up to see if you have had a chance to review our response to your comments. We would greatly appreciate any feedback regarding whether we have adequately addressed the concerns you raised in your review.
>
> Your insights are invaluable to us, and if you have any additional questions or suggestions, we would be more than happy to address them.
>
> Thank you once again for your time and thoughtful consideration.
>
> Best regards,
>
> All Authors

---

> ### Author Response · Authors · 2024-12-02
>
> Dear Reviewer,
>
> Since we are approaching the deadline of the discussion period, we were wondering if you have had the chance to review our response to your comments. **We also provided more details and case studies which may also address your concerns in our response to Reviewer kEed.**
>
> We would like to kindly inquire about the extent to which we have successfully addressed the concerns outlined in your review. We greatly value your feedback and would appreciate any further questions or comments you might have.
>
> Thank you for your time and consideration.
>
> Sincerely,
>
> All Authors

---

> > ### Comment · Reviewer_m95q · 2024-12-02
> >
> > Thanks for the reviewer's response, I've raised the scores accordingly.

---

> > > ### Author Response · Authors · 2024-12-02
> > >
> > > Thank you very much for recognizing our work!

---

### Official Review · Reviewer_Npyf · 2024-11-03

**Soundness:** 2
**Presentation:** 3
**Contribution:** 4
**Rating:** 8
**Confidence:** 4

**Summary:**

This paper introduces GasketRAG, a novel approach to improve the performance of Retrieval-Augmented Generation (RAG) systems.

The key contributions are:
1. A gasket model that acts as an intermediary between the retriever and the large language model (LLM) in the RAG pipeline. This gasket filters and refines the retrieved information to better align with both the retriever's and LLM's preferences.

2. A method for collecting high-quality preference data from both the LLM and retriever, which is used to train the gasket model offline. This avoids the complexity and instability of joint training approaches.

3. The use of weighted Kahneman-Tversky Optimization (KTO) to train the gasket model, which doesn't require initial supervised fine-tuning and enhances stability.

4. A two-iteration RAG process where the gasket model first filters initial retrieval results to enhance the query, then filters the results of a second retrieval before passing information to the LLM.

5. The authors evaluate GasketRAG against several baseline RAG methods across six datasets covering open-domain QA, multi-hop QA, and fact-checking tasks. Results show that GasketRAG outperforms existing methods on most test sets and metrics, demonstrating improved performance and stability across different question types.

6. The paper also highlights the fact about GasketRAG's ability to generalize, as the gasket model trained on preferences from one LLM (LLaMA-3-8B) still provides benefits when used with a different LLM (GPT-3.5-turbo) as the generator.

Overall, GasketRAG presents a novel approach to aligning retrievers and LLMs in RAG systems, addressing challenges in existing methods and demonstrating superior response accuracy and correctness.

**Strengths:**

**Originality:**
The paper presents a novel approach called GasketRAG that introduces an intermediate "gasket" model between the retriever and language model in retrieval-augmented generation (RAG) systems. This is an original idea that aims to systematically align the preferences of both components.


_Key original aspects include:_

1. Using a gasket model as an information bottleneck to control data flow and align the retriever and LLM
2. Collecting high-quality preference data from both the LLM and retriever to train the gasket offline
3. Employing a two-iteration RAG process with query enhancement
4. While building on existing work in RAG, this approach creatively combines ideas in a new way to address alignment challenges.


**Quality**
The paper demonstrates strong technical quality in several ways:
1. Rigorous experimental design comparing GasketRAG against multiple state-of-the-art baselines across 6 diverse datasets
2. Careful ablation studies examining the impact of different components by experimenting with different iteraton and weights.
3. Use of multiple evaluation metrics (Accuracy and Correctness) and comparing multiple baseline RAG systems.
4. Detailed analysis of results including failure cases such as how more iterations could be resulting in performance degradation, latency analysis.
5. Open-sourcing of code and data anonymosly to enable reproducibility
6. The use of weighted Kahneman-Tversky Optimization shows technical sophistication.
7. comparison of results between different setups  of the proposed approach such as effectiveness of weighted KTO.


**Clarity**
Overall, the paper is clearly written and well-structured. Key strengths in clarity include:

1. Clear problem motivation and positioning relative to prior work by citing multiple relevant recent research work.
2. Detailed method description with helpful diagrams (e.g. Fig 1 and 2) and logical division of implementations steps such as classifying the sentences list at high level first and then addressing each sentence group seperately.
3. Thorough experimental setup details with extnsive comarison between existing baselines.

Some sections (e.g. 3.3 Preference Collection) are quite dense and could potentially be clarified further. But on the whole, the paper effectively communicates the key ideas and contributions.

**Significance**
This work has the potential for significant impact in several ways:

1. Addressing a key challenge in RAG systems by better aligning retrievers and LLMs
2. Improving performance on important tasks like open-domain QA and fact-checking
3. Providing a flexible approach that can work with different off-the-shelf retrievers and LLMs
4. Offering insights into preference learning and information filtering for language models
5. The strong empirical results across multiple datasets suggest the approach could be widely applicable.
6. The ability to generalize to different LLMs (e.g. from LLaMA to GPT-3.5) is particularly noteworthy.

**Weaknesses:**

1. **Limited exploration of gasket model architectures:**  The paper uses LLaMA-3.1-8B-Instruct as the gasket model but does not explore alternative architectures or model sizes. Experimenting with smaller models or different architectures could provide insights into the trade-offs between performance and efficiency.
2. **Limited exploration of different retrievers:** The paper uses ColBERTv2 as the retriever but does not explore how GasketRAG performs with different retrieval methods. Because the retreiver place an important role in the proposed appraoch, experimenting with alternative retrievers would demonstrate the method's robustness and generalizability.
3. **Effective of various iteration scenarios against baseline:**  The effectiveness of iteration results claim that the 2-Iteration GasketRAG achieves the best overall performance but at the same time latency results shows that GasketRAG has slightly higher latency compared to SelfAsk and Iter-RetGen. However, the 1-Iteration Gasket is significantly faster than both while also delivering better performance. but there seems to be no experiments conducted for comparing GasketRAG with various iteration against other baseline RAG methods for accuracy and correctness to understand the impact of accuracy and correctness and make trade-offs.

**Questions:**

1. How does the runtime and resource usage of GasketRAG compare to baseline methods, especially for larger-scale applications?

2. Have the authors experimented with different architectures or model sizes for the gasket model besides LLaMA-3.1-8B-Instruct? What were the trade-offs observed between performance and efficiency when using smaller models?

3. Have the authors tested GasketRAG with different retrieval methods besides ColBERTv2?

4. Have the authors experimented different iterations mode of GasketRAG and compared Accuracy/Correctness with other base line RAG approaches along with its latency to demonstrate different trade-offs?

5. Have the authors identified any potential biases introduced by preference data collection method?

6. How well does GasketRAG perform in larger datasets?

---

> ### Author Response · Authors · 2024-11-16
>
> Thank you very much for your exceptionally thorough reading and understanding of our paper. We believe your feedback demonstrates a deep knowledge of the field, and we are more than happy to engage in a discussion on the issues you are concerned about.
>
> **Concern 1: Different gasket model architectures and resource usage.**
>
> **Response:**
>
> We additionally trained two gasket models based on Qwen-2.5: a 0.5B and a 1.5B model. These were compared with our gasket model based on LlaMA-3.1. The LLM generator is LLaMA-3-8B. The following are the test results (correctness):
>
> |  | 0.5B (Qwen-2.5) | 1.5B (Qwen-2.5) | 8B (LLaMA-3.1) |
> | --- | --- | --- | --- |
> | PopQA | 42.7 | 46.9 | 45.7 |
> | TriviaQA | 56.6 | 62.9 | 65.5 |
> | HotPotQA | 44.4 | 48.1 | 54.8 |
> | 2Wiki | 32.1 | 35.2 | 38.6 |
> | PubHealth | 73.0 | 72.4 | 72.1  |
> | StrategyQA | 57.0 | 57.5 | 58.1 |
>
> From the table, it can be observed that as the model size increases, the performance also shows improvement. The 0.5B model has performance limitations, indicating that regulating the retriever and LLM within the RAG pipeline requires the gasket model to possess a considerable level of text understanding. Additionally, it is evident that the 1.5B model demonstrates quite strong performance, slightly lagging behind the 8B model across datasets but still outperforming other RAG methods. This indicates that GasketRAG is **more efficient compared to other RAG methods**. By using a smaller model as the gasket model to filter sentences, the token sequence length input to the LLM generator is significantly reduced. As a result, despite requiring two iterations, GasketRAG has minimal resource demands.
>
> ---
>
> **Concern 2: Have the authors experimented different iterations mode of GasketRAG?**
>
> **Response:**
>
> We may not have fully understood the meaning of *different iterations mode* in this question. In Table 4, we present the performance of GasketRAG with varying numbers of iterations. A single iteration means that the gasket model acts merely as a refiner without adjusting the retriever. We observe that increasing the number of iterations to 3 results in a decline in stability compared to 2 iterations due to the accumulation of errors. Moreover, additional iterations exponentially increase resource consumption. Therefore, we consider two iterations to be the optimal choice, which aligns with our expectations.
>
> We also explored a variant of GasketRAG, where in each iteration, the sentence list output by the gasket model is passed to the LLM generator to produce a preliminary answer. This preliminary answer is then concatenated to the query and used as input for the next iteration's retriever. The test results (correctness) are as follows:
>
> |  | Vanilla Gasket | w/ Answer Aug. |
> | --- | --- | --- |
> | PopQA | 45.7 | 48.5 |
> | TriviaQA | 65.5 | 65.6 |
> | HotPotQA | 54.8 | 53.7 |
> | 2Wiki | 38.6 | 37.5 |
> | PubHealth | 72.1  | 72.6 |
> | StrategyQA | 58.1 | 57.8 |
>
> Using the answer to enhance the query in each iteration did not result in significant improvement. Instead, it increased the number of LLM generator calls, thereby consuming more resources. Considering that many RAG methods use draft answers to significantly improve the retriever's retrieval quality, GasketRAG can achieve similar effects efficiently through the gasket model, making it a more resource-efficient approach.

---

> > ### Comment · Reviewer_Npyf · 2024-11-16
> > **discrepancy between stability improvement and latency  for 2 iterations**
> >
> > The paper shows a discrepancy and it is also re-iterated in the comment above. if 2 number of iterations gives the most stability but at the same time paper proposes that  2 number of iterations decreases in latency, then  how this concept be proved for various scenarios of stability vs latency and especially in large scale use cases? Can you provide data points for the same?

---

> > > ### Author Response · Authors · 2024-11-17
> > >
> > > Thank you for your response and your willingness to continue the discussion with us.
> > >
> > > For the performance of GasketRAG with different numbers of iterations, we have presented the results in **Section 4.6**. GasketRAG with 2 iterations achieves the best performance, while increasing the number of iterations to 3 amplifies cumulative errors, leading to a performance decline.
> > >
> > > We have merged the data (correctness) from Table 2 and Table 4：
> > >
> > > |  | PopQA | TriviaQA | HotpotQA | WikiMultiHop |
> > > | --- | --- | --- | --- | --- |
> > > | NaiveRag | 43.7 | 61.0 | 47.5 | 32.0 |
> > > | RRR | 43.0 | 56.0 | 42.8 | 29.8 |
> > > | Iter-RetGen | 43.5 | 61.0 | 47.0 | 32.9 |
> > > | ActiveRAG | 44.6 | 60.9 | 47.3 | 30.1 |
> > > | SelfAsk | 16.6 | 36.1 | 30.2 | 26.9 |
> > > | SelfRAG | 37.4 | 50.8 | 38.4 | 22.1 |
> > > | 1-Iteration Gasket | 46.4 | 65.7 | 52.9 | 34.7 |
> > > | 2-Iteration Gasket | 45.7 | 65.5 | 54.8 | 38.6 |
> > > | 3-Iteration Gasket | 44.8 | 64.3 | 52.6 | 36.2 |
> > >
> > > The 2-iteration Gasket is the standard GasketRAG method presented in Table 2. It can be observed that the 1-iteration Gasket outperforms all other RAG methods while exhibiting significantly lower latency compared to SelfRAG, 2-iteration Gasket, SelfAsk, and Iter-RetGen (Figure 4).
> > >
> > > Increasing the number of iterations increases the calls to the gasket model, resulting in longer processing times. This does **not align with your interpretation that** **"*paper proposes that 2 number of iterations decreases in latency*".**
> > >
> > > We are not entirely sure what you mean by "***large-scale use cases.***" The latency of RAG methods is relatively straightforward to estimate, as it is directly proportional to the number of LLM calls, assuming the text length remains consistent. In GasketRAG, since the gasket model handles the longer text inputs while the generator processes only the question and selected sentences, the number of gasket model calls primarily determines the pipeline's latency.
> > >
> > > We hope our explanation addresses your concerns. If we have misunderstood your question, please correct us.

---

> > > > ### Comment · Reviewer_Npyf · 2024-12-02
> > > >
> > > > Let me try tor re-iterate to make it clear here. it is not clear that how may iterations have to be set to achieve best results with latency. if you can explain that and mention it in the paper, that would be useful for readers to better understand the effectiveness.
> > > >
> > > > by large scale use cases, i mean the size of the RAG database which might have PB scale of data where each iteration of query performance is critical.

---

> > > > > ### Author Response · Authors · 2024-12-03
> > > > >
> > > > > Thank you for your clarification. Considering the overall performance and latency, we believe that **2-Iteration GasketRAG is the optimal choice**. To this end, we extended the iteration comparison experiments by adding GasketRAG with 4-Its and 5-Its.
> > > > >
> > > > > |  | HotpotQA | WikiMultiHopQA | PopQA | TriviaQA |
> > > > > | --- | --- | --- | --- | --- |
> > > > > | 1-Iteration Gasket | 52.90 | 34.70 | 46.40 | 65.70 |
> > > > > | 2-Iteration Gasket | 54.80 | 38.60 | 45.70 | 65.50 |
> > > > > | 3-Iteration Gasket | 52.60 | 36.20 | 44.80 | 64.30 |
> > > > > | 4-Iteration Gasket | 54.90 | 38.30 | 48.10 | 66.30 |
> > > > > | 5-Iteration Gasket | 54.60 | 36.60 | 48.00 | 66.60 |
> > > > >
> > > > > It can be observed that with more iterations, the cumulative bias appearing in the third round is alleviated. However, the performance remains roughly equivalent to that of the 2-Iteration Gasket. Considering that the number of calls to the LLM and retriever increases proportionally with the number of iterations, 2 iterations is the best choice. We will make it clearer in the paper.
> > > > >
> > > > > In our experiments, we utilized the WiKi2018 corpus with 21 million passages, where the retriever's retrieval time was almost negligible compared to the LLM inference time (averaging below 0.1 seconds). Due to resource constraints, we were unable to validate our approach in PB-scale data scenarios. However, it is certain that GasketRAG with 2 retriever calls maintains an advantage over other iterative methods. In our experiments, Iter-RetGen required 3 calls, SelfAsk up to 5 calls, and SelfRAG up to 7 calls.
> > > > >
> > > > > Thank you for your active discussion and insightful feedback. We will be ready to address any of your questions right up until the very end.

---

> ### Author Response · Authors · 2024-11-16
>
> **Concern 3: The impact of preference dataset size.**
>
> **Response:**
>
> We used the same experimental setup and retrained the gasket model with half the preference data (8.5K). The results are as follows:
>
> |  | Gasket_8.5K | Gasket_17K |
> | --- | --- | --- |
> | PopQA | 48.1 | 45.7 |
> | TriviaQA | 65.0 | 65.5 |
> | HotPotQA | 52.1 | 54.8 |
> | 2Wiki | 37.3 | 38.6 |
> | PubHealth | 71.2 | 72.1  |
> | StrategyQA | 57.2 | 58.1 |
>
> The performance gap between the gasket model trained with 8.5K data and the one trained with 17K data is minimal, demonstrating the high data efficiency of our proposed preference data collection method and the Weighted KTO training algorithm. **GasketRAG requires only a small amount of preference data to achieve a high level of performance.** Since GasketRAG eliminates the need for pair-wise data, it does not require meticulously crafting large data combinations for contrastive learning.
>
> ---
>
> **Concern 4: Further study.**
>
> **Response:**
>
> - **Other retrievers**
>
>     We are currently replacing the ColBERTv2 retriever with others such as Contriever and replicate the experiments. It involves collecting new preference data, retraining the gasket model, and running evaluations. Due to computational resource limitations, this may take some time. We will include the results in the revised version once the process is complete.
>
> - **Preference data biases**
>
>     We are not entirely sure which aspect of bias you are referring to. So far, we have not specifically explored this area. We hope you can provide further insights or suggestions that might guide us in this direction.
>
>
> We hope our clarifications address your concerns and answer your questions. If there are any inaccuracies in our response, please feel free to point out. Thank you once again for your valuable feedback.

---

### Official Review · Reviewer_JX4G · 2024-11-04

**Soundness:** 2
**Presentation:** 2
**Contribution:** 2
**Rating:** 5
**Confidence:** 4

**Summary:**

This paper introduces a “gasket” component positioned between the retriever and the large language model (LLM) within the RAG (Retrieval-Augmented Generation) framework. This gasket is trained using the KTO algorithm to selectively filter relevant sentences from the retrieved passages, leveraging preference data collected from the LLM and retrieval models. Experiments conducted on six datasets across three tasks demonstrate that the proposed GasketRAG method outperforms prior RAG approaches in most cases. Detailed analyses of KTO training, iteration counts, and latency are also provided, highlighting the features and efficiency of the GasketRAG approach.

**Strengths:**

1) This paper proposes a new RAG pipeline, GasketRAG, which introduces a gasket model to regulate data flow for improved LLM generation.
2) Extensive experiments on six datasets across three tasks, along with detailed analyses, are conducted to evaluate the effectiveness of GasketRAG approach.

**Weaknesses:**

1) The title and motivation of this paper may be somewhat overstated, as the proposed GasketRAG lacks a clear alignment mechanism between the LLM and the retriever.
2) The proposed GasketRAG pipeline primarily relies on a model to filter relevant sentences and performs two retrieval steps before the LLM generation, which may lack sufficient novelty to make it stand out.
3) The description of preference data construction is somewhat disorganized, such as the settings of the base models for the gasket and generation components, as well as the distinctions between data construction and training process.

**Questions:**

1) How does the gasket model function as an aligner between large language models (LLMs) and retrieval models? My understanding is that the gasket is trained to select sentences that help the retriever find more relevant passages and assist the LLM in generating accurate answers. However, it’s unclear where the systematic alignment between the LLM and the retriever occurs.
2) How is “preference” defined for the LLMs and retrievers? According to Section 3.3, both LLM and retriever preferences are determined based on whether the LLM generator can produce a correct answer from the selected sentences. This seems to indicate that the “preference” signal comes solely from the LLMs. Besides, is it accurate to call this preference data?
3) Have you considered comparing the gasket model with a sentence-level ranking model, such as training a BERT model using the preference signal from the LLM to rank all sentences in the retrieved passages? This approach could potentially offer greater efficiency than an LLM-based filtering model, as BERT is generally faster and more lightweight.
4) In lines 201–203, it states “In the preferred group, for each y′, we calculate the average sentence index number. A lower average index number indicates that the retriever ranks useful information higher.” Why is the sentence index linked to retrieval effectiveness? The second-round retrieval yields a new set of passages, and how to assign the sentence IDs to these passages?

---

> ### Author Response · Authors · 2024-11-16
>
> Thank you for your thoughtful review and feedback on our submission. We appreciate the time and effort you invested in understanding our work. We would like to address the concerns you raised:
>
> **Concern 1: Alignment mechanism in GasketRAG and preference definition.**
>
> **Response:**
>
> We understand that you believe the gasket model simply selects relevant sentences to help the retriever find more pertinent documents and assist the LLM in generating responses. However, the issue lies in how "relevance" is defined. Retrievers are generally trained based on human preferences, designed to retrieve and rank documents in a way that aligns with human habits. However, the preferences of LLMs do not completely align with those of humans.
>
> As pointed out in [1], when LLMs are presented with both LLM-generated and retrieved documents, they tend to favor content generated by LLMs. This demonstrates subtle differences in how LLMs process contexts from different sources. Furthermore, [2] validates that LLMs are not particularly sensitive to the ranking of documents but are highly sensitive to irrelevant information (noise) within documents. **We define the divergence between the retriever's output preferences and the LLM's input preferences as the *preference gap*.**
>
> Thus, the gasket model does more than merely select relevant sentences. **Its underlying logic involves manipulating the content and noise in the inputs to both the retriever and the LLM to modulate their behavior and bridge this preference gap.** For example, for certain queries, an LLM might possess parameter knowledge and can provide answers without requiring additional context. In such cases, the content retrieved by the retriever might have a negative impact. The gasket model learns these preferences from training data and tends to avoid selecting any sentence from the retrieved passages to reduce interference when it believes the LLM is able to answer without assistance.
>
> Of course, the preferences of LLMs and retrievers regarding input content are complex and are treated as a black box in our approach. By focusing on the final answer as the goal, the gasket model coordinates all components within the RAG pipeline. This coordination involves simultaneously controlling the inputs to both the retriever and the LLM. We refer to this process as ***systematic alignment***.
>
> [1] Hexiang Tan, Fei Sun, Wanli Yang, Yuanzhuo Wang, Qi Cao, and Xueqi Cheng. 2024. Blinded by Generated Contexts: How Language Models Merge Generated and Retrieved Contexts When Knowledge Conflicts?
>
> [2] Zixuan Ke, Weize Kong, Cheng Li, Mingyang Zhang, Qiaozhu Mei, and Michael Bendersky. 2024. Bridging the Preference Gap between Retrievers and LLMs.
>
> ---
>
> **Concern 2: Novelty of our method.**
>
> **Response:**
>
> Existing methods typically focus on specific components of the RAG pipeline. For instance, some use refinement techniques to help the LLM better understand the content returned by the retriever, while others employ question rewriting to improve the retriever's relevance in retrieving documents. While these approaches have achieved certain improvements, they often overlook the *preference gap* between the LLM and the retriever. Bridging this gap offers additional potential for enhancement.
>
> The traditional solution to this issue involves jointly training the LLM and the retriever using algorithms such as PPO, with reward models incorporated. However, this significantly increases the complexity and instability of the framework, limiting the scope of research in this area.
>
> The innovation of our approach lies in the fact that it does **not rely on human annotations** (which, in fact, may fail to align with the preference gap) **nor on building a complex joint training framework** involving all components of the RAG pipeline. Instead, our method introduces a **straightforward and effective way to collect preference data** and utilizes a **Weighted KTO** method. Our experiments demonstrate that GasketRAG outperforms iterative retrieval, chain-of-thought (CoT) reasoning, and question rewriting methods in terms of performance and exhibits **greater stability** across different datasets and various LLMs.

---

> ### Author Response · Authors · 2024-11-16
>
> **Concern 3: Using a lightweight BERT ranking model.**
>
> **Response:**
>
> You’ve brought up an excellent perspective. First, we should point out that training BERT using preference data is actually more complex because BERT is an encoder-only model. To perform sentence-level ranking, it requires constructing a pair-wise training dataset for contrastive learning.
>
> The advantage of our method lies in the simplicity of collecting preference data. It only requires labeling each record as *preferred* or *dispreferred* (with strong/weak weights), without needing pairwise comparisons or considering their distribution in the embedding space. This makes our approach more **data-efficient** and **stable**.
>
> Additionally, as previously mentioned, LLMs are not particularly sensitive to document ranking. Therefore, what the gasket model does is fundamentally different from re-ranking. Instead of simply reordering, it edits the inputs to both the retriever and the LLM, enabling a more systematic alignment.
>
> For efficiency concerns, we trained additional lightweight gasket models based on Qwen2.5 0.5B and 1.5B. Following is the correctness of the models tested on various datasets:
>
> |  | 0.5B (Qwen-2.5) | 1.5B (Qwen-2.5) | 8B (LLaMA-3.1) |
> | --- | --- | --- | --- |
> | PopQA | 42.7 | 46.9 | 45.7 |
> | TriviaQA | 56.6 | 62.9 | 65.5 |
> | HotPotQA | 44.4 | 48.1 | 54.8 |
> | 2Wiki | 32.1 | 35.2 | 38.6 |
> | PubHealth | 73.0 | 72.4 | 72.1  |
> | StrategyQA | 57.0 | 57.5 | 58.1 |
>
> From the table, it can be observed that the 0.5B model has performance limitations, indicating that regulating the retriever and LLM within the RAG pipeline requires the gasket model to possess a considerable level of text understanding. Additionally, it is evident that the 1.5B model demonstrates quite strong performance, slightly lagging behind the 8B model across datasets but still outperforming other RAG methods. This indicates that GasketRAG is more efficient compared to other RAG methods. By using a smaller model as the gasket model to filter sentences, the token sequence length input to the LLM generator is significantly reduced. As a result, despite requiring two iterations, GasketRAG has minimal resource demands.
>
> ---
>
> **Concern 4: How is the sentence index linked to retrieval effectiveness?**
>
> **Response:**
>
> In the first iteration, the sentences selected by the gasket model are used to strengthen the query, influencing the retriever’s tendency in the second round of retrieval. We hypothesize that if the useful information in the retriever's results is ranked closer to the top in the second round, it indicates that the sentences selected by the gasket model are of higher quality. Consequently, if the average index of the sentences chosen by the gasket model in the second round decreases, it implies that useful information is being prioritized in the retrieval results. This brings two benefits:
>
> 1. **Preventing Cheating through Redundant Selection**: By ensuring that the gasket model selects relevant sentences while avoiding irrelevant ones, it prevents the model from "cheating" by indiscriminately copying large portions of retrieved documents. This improves its ability to distinguish between *golden sentences* (highly relevant) and *silver sentences* (less relevant).
> 2. **Promoting Discovery of Potentially Useful Documents**: The overall forward shift of useful information implies that documents that were previously not included in the retrieval results, but could be relevant, are now promoted to appear in the second round's results.
>
> To illustrate this process more concretely, consider the following example: Suppose all retrieved documents are concatenated as follows:
>
> ```
> [1] Title: Passage 1. [2] Sentence... [3] Title: Passage 2. [4] Sentence... [5] Sentence...
> ```
>
> After the first round of retrieval, the gasket model selects sentences [2] and [4]**.** We calculate their average index as (2+4)/2=3. Using these sentences to enhance the query for the second round of retrieval, a new batch of documents is retrieved. As before, sentences are renumbered independently of the first round. Suppose the gasket model then selects a new set of sentences [1], [2], and [4]. We calculate the new average index as: (1 + 2 + 4)/3=2.33. The decrease in the average index indicates that the retriever has successfully prioritized relevant information in the second round.
>
> We hope our clarifications address your concerns and answer your questions. Thank you once again for your valuable insights.

---

> ### Author Response · Authors · 2024-11-26
>
> Dear reviewer,
>
> I hope this message finds you well.
>
> We wanted to kindly follow up to see if you have had a chance to review our response to your comments. We would greatly appreciate any feedback regarding whether we have adequately addressed the concerns you raised in your review.
>
> Your insights are invaluable to us, and if you have any additional questions or suggestions, we would be more than happy to address them.
>
> Thank you once again for your time and thoughtful consideration.
>
> Best regards,
>
> All Authors

---

> > ### Comment · Reviewer_JX4G · 2024-11-28
> >
> > Thank you for your detailed response, which clarified some of my concerns. However, I still think “the systematic alignment of large language models and retrievers” somewhat overstated, as the fundamental supervision signals are solely based on whether the LLM can answer questions correctly, rather than influencing or accounting for the retriever’s preferences—whether for AI-generated text, human-written content, or other textual signals. Meanwhile, considering the concerns raised by other reviewers, such as limited improvement and baselines, I will keep my score unchanged.

---

> > > ### Author Response · Authors · 2024-11-28
> > >
> > > Thank you for your response.
> > >
> > > Regarding our title, the word *systematic* is defined in the Oxford Languages dictionary as "done or acting according to a fixed plan or system." We use this term in our title to convey that GasketRAG thoroughly considers the preferences of both the retriever and the LLM. Therefore, we are unsure why you perceive this as an overclaim. All the contributions we listed in the paper are supported by corresponding experiments.
> > >
> > > As for *GasketRAG based on whether the LLM can answer questions correctly*, ensuring that the LLM provides correct answers is the ultimate goal of any RAG pipeline. Consequently, we align the Gasket model's control over the retriever and LLM with this goal, offering an efficient and practically valuable training approach. Through our preference data collection method, the Gasket model implicitly learns retriever preferences. In our responses to reviewers m95q and kEed, we provided case studies demonstrating how the Gasket model leverages learned preferences to control the retriever's behavior. Reviewer kEed had raised concerns about retriever preferences, and our response effectively addressed these concerns, as evident from their feedback.
> > >
> > > Regarding the baseline concerns raised by reviewer UHMc, we provided clarifications and additional experimental results. Their feedback indicates that our explanations successfully resolved their doubts in this area. Additionally, reviewer Npyf expressed positive recognition of our experimental results in their review. GasketRAG significantly outperformed all other RAG methods on five out of six tested datasets and maintained high stability on all of the datasets, even in cases where other methods performed poorly on specific datasets. In the results of w/ GPT-3.5 in Table 2, we used the same gasket model aligned with LLaMA-3, rather than realigning it with GPT-3.5. Our method still demonstrated substantial improvement over other RAG methods. Reviewer Npyf also noted that *the ability to generalize to different LLMs is particularly noteworthy.* Furthermore, comparative experiments using a 1.5B smaller model and less training data highlighted our approach's superiority in computational and data efficiency. Therefore, the improvements brought by our method are not limited.
> > >
> > >
> > > We hope that our clarifications can resolve any misunderstandings, and we kindly request that you reconsider your evaluation of our work. Thank you once again for taking the time to review our response.

---

> > > > ### Comment · Reviewer_JX4G · 2024-12-03
> > > >
> > > > Thanks for the rebuttal. I still have the following questions:
> > > > 1) How do you define the preferences of the retriever and the LLM? The Gasket model selects sentences that enable the retriever (in the second iteration) to retrieve more relevant ones, resembling a pseudo-relevance feedback mechanism [1] commonly used in information retrieval. However, I am curious about why this is considered a “preference” of the retriever, as it seems more like an optimization goal rather than an inherent preference. Could you clarify this distinction?
> > > > 2) I noticed that the paper [2] also seeks to bridge the preference gap between the retriever and the LLM, adopting a multi-stage training process to develop a Bridge model similar to the Gasket model described in your paper. Could you elaborate on the key differences between these two approaches? Additionally, why wasn’t the BGM model included in the experimental comparisons? Was it due to specific constraints, differences in methodology, or another reason?
> > > >
> > > > [1] How Does Feedback Signal Quality Impact Effectiveness of Pseudo Relevance Feedback for Passage Retrieval. In SIGIR 2022.
> > > >
> > > > [2]  Bridging the Preference Gap between Retrievers and LLMs. In ACL 2024.

---

> > > > > ### Author Response · Authors · 2024-12-04
> > > > >
> > > > > Thank you for your question. The preferences of the retriever and LLM refer to their focus on specific aspects of the input information and their sensitivity to local context. For the retriever preferences, when given a query the retriever might exhibit varying sensitivity to different information points in the question or might misinterpret the query semantics. As a result, the retrieved documents reflect the "preference (or bias)" of the retriever.
> > > > >
> > > > > The Gasket model leverages preference learning to utilize or correct such preferences (or biases) with the goal of aligning them with the preferences of the LLM. This alignment ensures that the retriever's output is better understood and utilized by the LLM.
> > > > >
> > > > > The optimization goal and preference are not mutually exclusive concepts, as preference learning is essentially a form of optimization. Since GasketRAG employs preference learning algorithms and collects data and feedback from both the retriever and LLM components, the Gasket model aligns the preferences of the retriever with those of the LLM. The feedback from the LLM is relatively easy to understand, as it is based on the correctness of the LLM's responses. **In contrast, the feedback from the retriever comes from the average position of the positive sample sentences, selected by the Gasket model, within the retrieval results.**
> > > > >
> > > > > You are attempting to understand the retriever preference learning component of GasketRAG within the framework of the pseudo-relevance feedback mechanism [1]. We do not reject this approach, nor do we have any intention of dissociating our method from pseudo-relevance feedback.  What we want to emphasize is how we achieve this alignment in a simple and efficient manner within the RAG framework. Previous methods have attempted to achieve alignment in the RAG framework by either independently training the retriever or the LLM, or by jointly training both. However, these approaches often require substantial amounts of data and computational resources, while also lacking flexibility. In contrast, GasketRAG integrates modifications for both the retriever and LLM into a single gasket model. This design allows for seamless replacement of any retriever or LLM, requiring only adjustments to the gasket model, thereby making the process much more efficient and adaptable.
> > > > >
> > > > > In summary, the novelty of our approach lies in:
> > > > >
> > > > > 1. **Simultaneous Alignment Through the Gasket Model**: Achieving alignment between the retriever and LLM by jointly controlling both components via the gasket model.
> > > > > 2. **Efficient Preference Data Collection and Training**: Introducing an effective methodology for collecting preference data and training the model, significantly reducing resource requirements while maintaining flexibility.
> > > > >
> > > > > ---
> > > > >
> > > > > The differences between GasketRAG and the BGM method [2] are as follows:
> > > > >
> > > > > 1. **Finer-Grained Control**: GasketRAG achieves more granular control by regulating the information flow between the retriever and LLM at the sentence level via the gasket model. This results in a narrower information bottleneck, leading to more efficient model training.
> > > > > 2. **Enhanced Query Iteration**: Unlike BGM, where outputs are based on passages, the gasket model in GasketRAG outputs sentences. These sentence-level outputs can be used to enhance the query, enabling iterative improvements in the retriever's results.
> > > > > 3. **Simplified Training Process**: BGM employs a two-stage training process (Supervised Learning + Reinforcement Learning), whereas GasketRAG uses a single-stage offline training process. This significantly reduces complexity, computational requirements, and data volume needed for effective training.
> > > > >
> > > > > Since BGM has not released its model or implementation details, we cannot directly compare our approach with theirs. However, the 1-Iteration GasketRAG can be considered a method analogous to BGM. When GasketRAG does not enhance the query and simply filters and reranks the retriever's results, it aligns with the functionality of the BGM method. Additionally, by operating at the sentence level, the gasket model offers superior noise reduction capabilities.
> > > > >
> > > > > Our experiments, illustrated in Figure 3, indicate that supervised SFT does not yield optimal results for gasket model training. Moreover, extensive practical evidence demonstrates that KTO training provides greater stability compared to PPO training. Consequently, the 1-Iteration GasketRAG is also a strong baseline.
> > > > >
> > > > > Table 4 highlights the performance gap between 1-Iteration GasketRAG and 2-Iteration GasketRAG (standard). These results show that relying solely on one-way alignment from retriever to LLM is insufficient for achieving optimal outcomes. By editing the query to refine the retriever's search results, the preference gap between the LLM and retriever can be further minimized.
> > > > >
> > > > > Thank you again for your thorough review. We hope our responses have addressed your concerns effectively.

---

### Author Response · Authors · 2024-11-23
**New Version Uploaded**

Dear reviewers,

We appreciate your feedback on our work and the time and effort you dedicated to it. Based on the feedback from all reviewers, **we have submitted a new version of the PDF**, which includes the following revisions:

1. Improved the articulation of the motivation, clearly highlighting our contributions.
2. Added additional experiments, including the use of more architecture models, comparisons with different training data sizes, comparisons with SFT training, etc.

**The modifications are highlighted in red font.**

**We kindly ask if you've had a chance to review our response to your comments. We'd appreciate knowing if we've adequately addressed your concerns and any further questions you might have.**

Thank you again for your time.

Sincerely,

All Authors

---

### Note · Authors · 2025-01-23

I have read and agree with the venue's withdrawal policy on behalf of myself and my co-authors.